# Endothelial Ca²⁺ oscillations reflect VEGFR signaling-regulated angiogenic capacity in vivo

Yasuhiro Yokota[1†], Hiroyuki Nakajima[1†], Yuki Wakayama[1], Akira Muto[2,3], Koichi Kawakami[2,3], Shigetomo Fukuhara[1], Naoki Mochizuki[1,4*]

[1]Department of Cell Biology, National Cerebral and Cardiovascular Center Research Institute, Suita, Japan; [2]Division of Molecular and Developmental Biology, National Institute of Genetics, Mishima, Japan; [3]Department of Genetics, SOKENDAI (The Graduate University for Advanced Studies), National Institute of Genetics, Mishima, Japan; [4]AMED-CREST, Japan Agency for Medical Research and Development, Suita, Japan

*For correspondence: nmochizu@ri.ncvc.go.jp

[†]These authors contributed equally to this work

Competing interests: The authors declare that no competing interests exist.

**Abstract** Sprouting angiogenesis is a well-coordinated process controlled by multiple extracellular inputs, including vascular endothelial growth factor (VEGF). However, little is known about when and how individual endothelial cell (EC) responds to angiogenic inputs in vivo. Here, we visualized endothelial Ca²⁺ dynamics in zebrafish and found that intracellular Ca²⁺ oscillations occurred in ECs exhibiting angiogenic behavior. Ca²⁺ oscillations depended upon VEGF receptor-2 (Vegfr2) and Vegfr3 in ECs budding from the dorsal aorta (DA) and posterior cardinal vein, respectively. Thus, visualizing Ca²⁺ oscillations allowed us to monitor EC responses to angiogenic cues. Vegfr-dependent Ca²⁺ oscillations occurred in migrating tip cells as well as stalk cells budding from the DA. We investigated how Dll4/Notch signaling regulates endothelial Ca²⁺ oscillations and found that it was required for the selection of single stalk cell as well as tip cell. Thus, we captured spatio-temporal Ca²⁺ dynamics during sprouting angiogenesis, as a result of cellular responses to angiogenic inputs.

## Introduction

An extensive branched blood vessel network is crucial to deliver oxygen and nutrients to tissues and organs. Such branched structures of blood vessels form mainly by sprouting angiogenesis, which involves the emergence of new vessels from the pre-existing vasculature (*Eilken and Adams, 2010*; *Herbert and Stainier, 2011*; *Phng and Gerhardt, 2009*). In the early steps of sprouting angiogenesis, specific motile endothelial cells (ECs) activated by angiogenic cues become leading tip cells and migrate outward from the parental vessels (*Gerhardt et al., 2003*). As the tip cell migrates from the parental vessel, neighboring ECs migrate by following the tip cell as stalk cells to keep connection between the tip cell and the parental vessel. While tip cells guide the sprouts, stalk cells constitute the base of the sprouts and are, therefore, considered to be less active than tip cells.

Sprouting angiogenesis is triggered by extracellular stimuli. Among them, vascular endothelial growth factor (VEGF)-A induces EC motility and loosen interendothelial cell junction by activating a tyrosine kinase receptor, VEGF receptor-2 (VEGFR2), to induce sprouting (*Ferrara, 2009*; *Lohela et al., 2009*). VEGF-A stimulates proliferation of ECs as well as survival (*Koch and Claesson-Welsh, 2012*; *Lohela et al., 2009*). VEGFR3 (also known as Flt4), a receptor of VEGF-C, is also required for developmental or tumor angiogenesis (*Tammela et al., 2008*). Especially, Vegfr3 is essential for venous sprouting from the posterior cardinal vein (PCV) in zebrafish (*Hogan et al.,*

**eLife digest** Throughout life, new blood vessels grow out like branches from existing vessels in a process called "sprouting angiogenesis". This involves some of the endothelial cells that line the inner surface of the blood vessel migrating outwards, creating a vessel sprout made up of tip cells and stalk cells.

Sprouting is controlled by two opposing signaling systems. One pathway is triggered by a molecule called vascular endothelial growth factor (VEGF). This molecule binds to receptor proteins to activate a range of signaling processes that stimulate endothelial cells to become tip cells, and so encourage the formation of new sprouts. However, it was not known exactly when or how the endothelial cells respond to these signals.

By contrast, the Notch signaling pathway inhibits sprouting angiogenesis. The two signaling pathways interact with each other: VEGF signaling in tip cells activates Notch signaling in neighboring cells, which then prevents VEGF signaling in these cells. This feedback mechanism helps a new sprout to form by suppressing tip-like activity in the cells surrounding a new tip cell, forcing these cells to become stalk cells.

Activating VEGF receptors also causes brief increases, or oscillations, in the level of calcium ions inside the endothelial cells. Now, Yokota, Nakajima et al. have investigated VEGF activity by genetically engineering zebrafish embryos so that fluorescent proteins inside their endothelial cells emit more light when calcium ion levels inside the cell increase. As zebrafish embryos are transparent, this change in fluorescence can be seen in the living animal. Imaging the embryos revealed that calcium ion oscillations occur in both tip and stalk cells in response to VEGF signaling as they bud from vessels. Notch signaling can also regulate the calcium ion oscillations; this controls whether an individual cell becomes a tip or a stalk cell, and restricts the number of stalk cells in the sprout.

The flow of blood through the vessels is also thought to influence calcium ion oscillations in endothelial cells. Future studies could therefore use the imaging technique developed by Yokota, Nakajima et al. to investigate how blood flow influences the development of new blood vessels.

*2009*). However, little is known about how VEGF-A/VEGFR2 or VEGF-C/VEGFR3 signaling in ECs induces initial sprouting in vivo.

While sprouting angiogenesis is promoted by VEGFR, it is negatively coordinated by delta-like 4 (Dll4)/Notch signaling (*Eilken and Adams, 2010*; *Lohela et al., 2009*; *Phng and Gerhardt, 2009*). During sprouting angiogenesis, Dll4/Notch signaling between ECs determines the selection of single tip cells among ECs of the pre-existing vessel by restricting the angiogenic behavior of neighboring ECs (*Hellström, et al., 2007*). In studies using mouse and zebrafish, loss of Dll4/Notch signaling results in an increased number of ECs showing tip cell behavior (*Hellström, et al., 2007*; *Leslie et al., 2007*; *Siekmann and Lawson, 2007*; *Suchting et al., 2007*). While VEGF-A/VEGFR2 signaling and Dll4/Notch signaling have opposing roles in regulating blood vessel sprouting, they are tightly linked in a negative feedback loop. VEGFR2 activation in tip cells leads to an up-regulation of the Notch ligand Dll4 and subsequent activation of Notch signaling in the following stalk cells (*Lobov et al., 2007*; *Suchting et al., 2007*; *Zarkada et al., 2015*). Increased Notch activity leads to the downregulation of VEGFR2 and/or VEGFR3, thereby restricting VEGFR2- or VEGFR3-dependent signaling in the stalk cells (*Benedito et al., 2012*; *Siekmann and Lawson, 2007*; *Zarkada et al., 2015*). Indeed, VEGFR3 expression is suppressed by Dll4/Notch signaling in zebrafish and mice (*Benedito et al., 2012*; *Siekmann and Lawson, 2007*; *Tammela et al., 2008*; *Zygmunt et al., 2011*). On the other hand, it is unclear to what extent VEGFR2 expression is suppressed by Dll4/ Notch signaling (*Benedito et al., 2012*; *Jakobsson et al., 2010*; *Suchting et al., 2007*). VEGFR2 and VEGFR3 are responsible for the aberrant angiogenesis induced by loss of Dll4/Notch signaling (*Hogan et al., 2009*; *Leslie et al., 2007*; *Siekmann and Lawson, 2007*; *Suchting et al., 2007*; *Zarkada et al., 2015*); however, it is still unclear how ECs behave in response to VEGFR-dependent angiogenic cues and how their behavior is spatio-temporally regulated by Dll4/Notch signaling.

Binding of VEGF-A to VEGFR2 induces dimerization of VEGFR2 and autophosphorylation of tyrosine residues in VEGFR2, leading to the activation of various downstream signaling pathways.

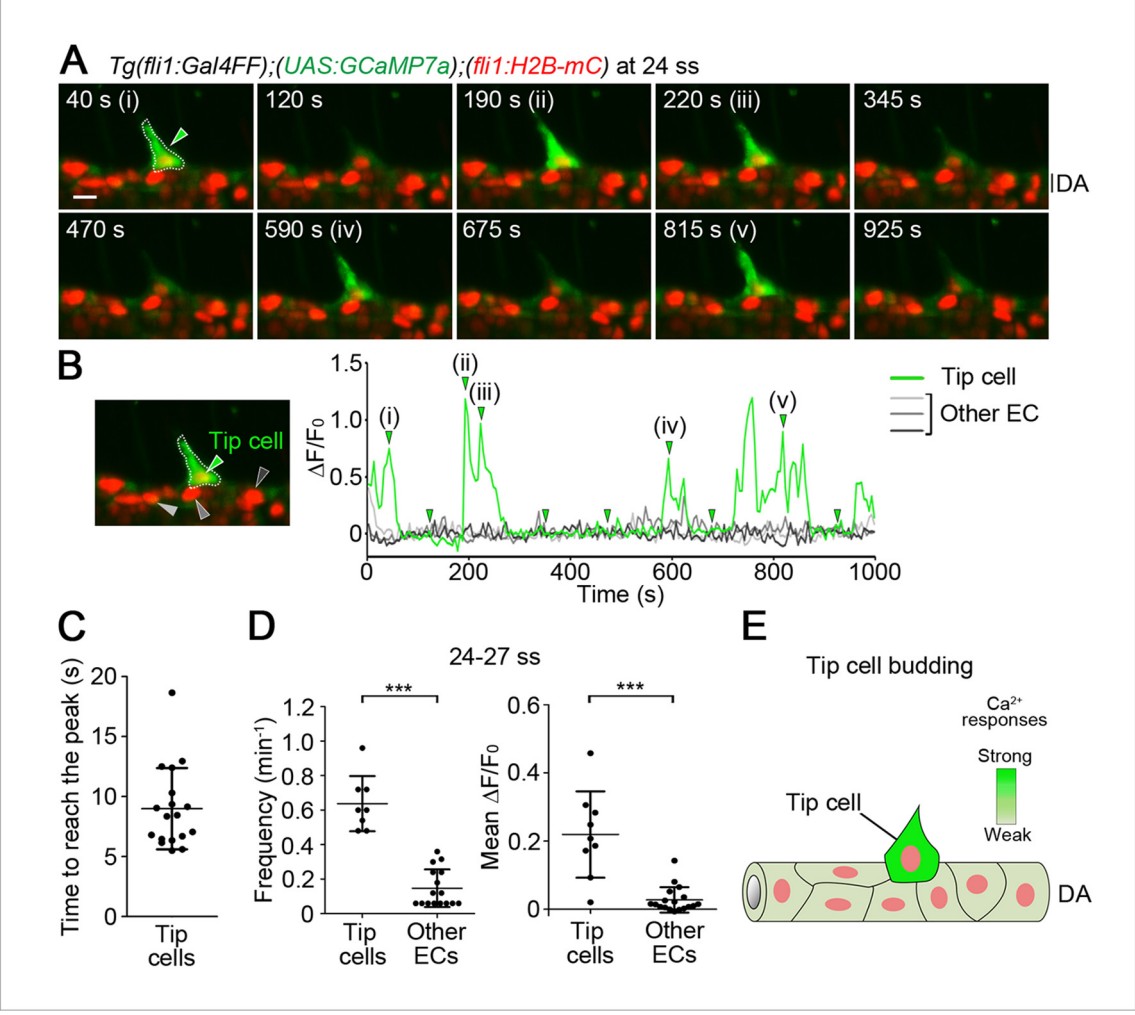

**Figure 1.** $Ca^{2+}$ oscillations in tip cells during budding from the dorsal aorta (DA). (**A**) 3D-rendered time-sequential images of the trunk regions of *Tg (fli1:Gal4FF);(UAS:GCaMP7a);(fli1:H2B-mC)* embryos during vessel sprouting from the DA (24 somite stage (ss)). 3D images were acquired using a light sheet microscope. The merged images of GCaMP7a (green) and H2B-mC (red) images are shown in the following images, unless otherwise described. All the zebrafish images are lateral views and displayed as anterior to the left. A green arrowhead indicates a tip cell outlined by a dashed line. (i)-(v) and other images are those indicated by the arrowheads indicated at a graph in **B**. (**B**) The fluorescence changes in GCaMP7a ($\Delta F/F_0$) of individual ECs from **A** indicated by arrowheads (green, light gray, dark gray, and black) at the left panel are shown as a graph. To measure the fluorescence intensity of GCaMP7a (green) in individual EC, the cell nucleus (red) was tracked over time (see 'Materials and methods'). (**C**) Dot-plot graphs depicting the time to reach the peak of each $Ca^{2+}$-oscillation in tip cells. Time-lapse 2D slice images of *Tg(fli1:Gal4FF);(UAS:GCaMP7a);(fli1:H2B-mC)* embryos taken every 100 ms as in *Figure 1—figure supplement 2B* were analyzed for quantification. Horizontal lines represent mean ± s.d. (n = 18). (**D**) Quantification of $Ca^{2+}$ oscillation frequency (left) and mean $\Delta F/F_0$ (right) in tip cells and other ECs within the DA during tip cell budding (24–27 ss) (see 'Materials and methods'). Each dot represents the value for a single cell. Horizontal lines represent mean ± s.d. (n ≥ 8). (**E**) Schematic model of tip cells showing $Ca^{2+}$ oscillations when they sprout from the DA. Intensity of green reflects the frequency of $Ca^{2+}$ oscillations. Scale bar, 10 µm in A. ***p < 0.001. DA, dorsal aorta.

The following figure supplements are available for figure 1:

**Figure supplement 1.** GCaMP7a works as a $Ca^{2+}$ indicator in endothelial cells (ECs).

**Figure supplement 2.** Quantitative analyses of intracellular $Ca^{2+}$ dynamics in ECs.

Autophosphorylated VEGFR2 associates with phospholipase C-γ (PLCγ) and increases intracellular $Ca^{2+}$ via accumulation of inositol 1,4,5-triphosphate ($IP_3$) (***Koch and Claesson-Welsh, 2012***;

*Moccia et al., 2012*) and store-operated Ca$^{2+}$ entry (*Li et al., 2011*). Besides VEGF-A, various chemical and mechanical stimuli induce an increase of intracellular Ca$^{2+}$ as a common second messenger in ECs (*Ando and Yamamoto, 2013*; *Moccia et al., 2012*). However, little is known about when and where the Ca$^{2+}$ responses in ECs occur during vascular development in vivo. We considered that we could precisely monitor Ca$^{2+}$ responses in vivo by using genetically encoded fluorescent Ca$^{2+}$ monitoring probes which can be used for detecting rapid Ca$^{2+}$ responses in living animals (*Muto et al., 2013*; *Rose et al., 2014*). We hypothesized that Ca$^{2+}$ dynamics might represent one of EC responses to angiogenic cues during sprouting angiogenesis.

In this study, we succeeded in visualizing the intracellular Ca$^{2+}$ dynamics in ECs at single-cell resolution in zebrafish by performing high-speed, three-dimensional (3D) time-lapse imaging, and uncovered how Ca$^{2+}$ dynamics are spatio-temporally regulated during sprouting angiogenesis. Intracellular Ca$^{2+}$ oscillations occurred in migrating tip and stalk cells, in a manner dependent upon VEGFR signaling. By investigating when and how Dll4/Notch signaling regulates endothelial Ca$^{2+}$ oscillations, we demonstrated how suppressive Dll4/Notch signaling was involved in the selection of tip cells from the dorsal aorta (DA) and that of stalk sells in intersomitic vessels (ISVs).

## Results

### Ca$^{2+}$ oscillations in budding tip cells from the DA

To understand how individual EC responds to angiogenic stimuli, we examined the dynamics of intracellular Ca$^{2+}$ in ECs during sprouting angiogenesis in vivo. We conducted in vivo Ca$^{2+}$ imaging by expressing a genetically encoded Ca$^{2+}$ indicator, GCaMP7a (GFP-based Ca$^{2+}$ probe) (*Muto et al., 2013*) in ECs of zebrafish. GCaMP7a is an improved version of GCaMP (*Nakai et al., 2001*), an engineered GFP that increases fluorescence upon the Ca$^{2+}$ elevation (*Figure 1—figure supplement 1A*). Firstly, we established a transgenic fish line, *Tg(fli1:Gal4FF);(UAS:GCaMP7a)*, in which GCaMP7a was driven by the endothelial specific promoter *fli1* via the Gal4/UAS system (*Asakawa et al., 2008*). This Tg line showed an increase of fluorescence exclusively in ECs in response to Ca$^{2+}$ elevation (*Figure 1—figure supplement 1B*). Secondly, to distinguish each EC, we developed a Tg fish line, *Tg(fli1:H2B-mC)*, in which EC nuclei was labeled by H2B-mCherry, and crossed this line with the *Tg(fli1:Gal4FF);(UAS:GCaMP7a)* line. We confirmed that almost all ECs expressed GCaMP7a in developing trunk vessels of these triple Tg embryos (*Figure 1—figure supplement 2A*), although the expression of GCaMP7a varied among ECs. To monitor fast Ca$^{2+}$ dynamics in ECs (see *Figure 1—figure supplement 2B,C*), we used a light sheet microscopy, which allows rapid acquisitions in living embryos by illuminating the sample with a focused light sheet perpendicularly to the direction of observation (*Huisken et al., 2004*).

We examined intracellular Ca$^{2+}$ dynamics in budding ECs of the DA close to somite boundaries at 24–27 somite stages (ss). We defined these budding ECs as tip cells, because we confirmed that they eventually became tip cells. These tip cells showed sustained and non-periodic Ca$^{2+}$ oscillations

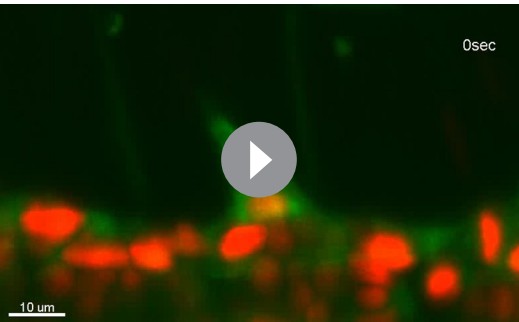

**Video 1.** Ca$^{2+}$ oscillations in tip cell during budding from the dorsal aorta (DA). Time-lapse recording of 3D-rendered light sheet images of the *Tg(fli1:Gal4FF);(UAS:GCaMP7a);(fli1:H2B-mC)* embryos at 24 somite stage (ss). Green, GCaMP7a fluorescence; red, H2B-mC fluorescence. Elapsed time from the start point of imaging is in seconds (s). Lateral view, anterior to the left. Scale bar, 10 μm.

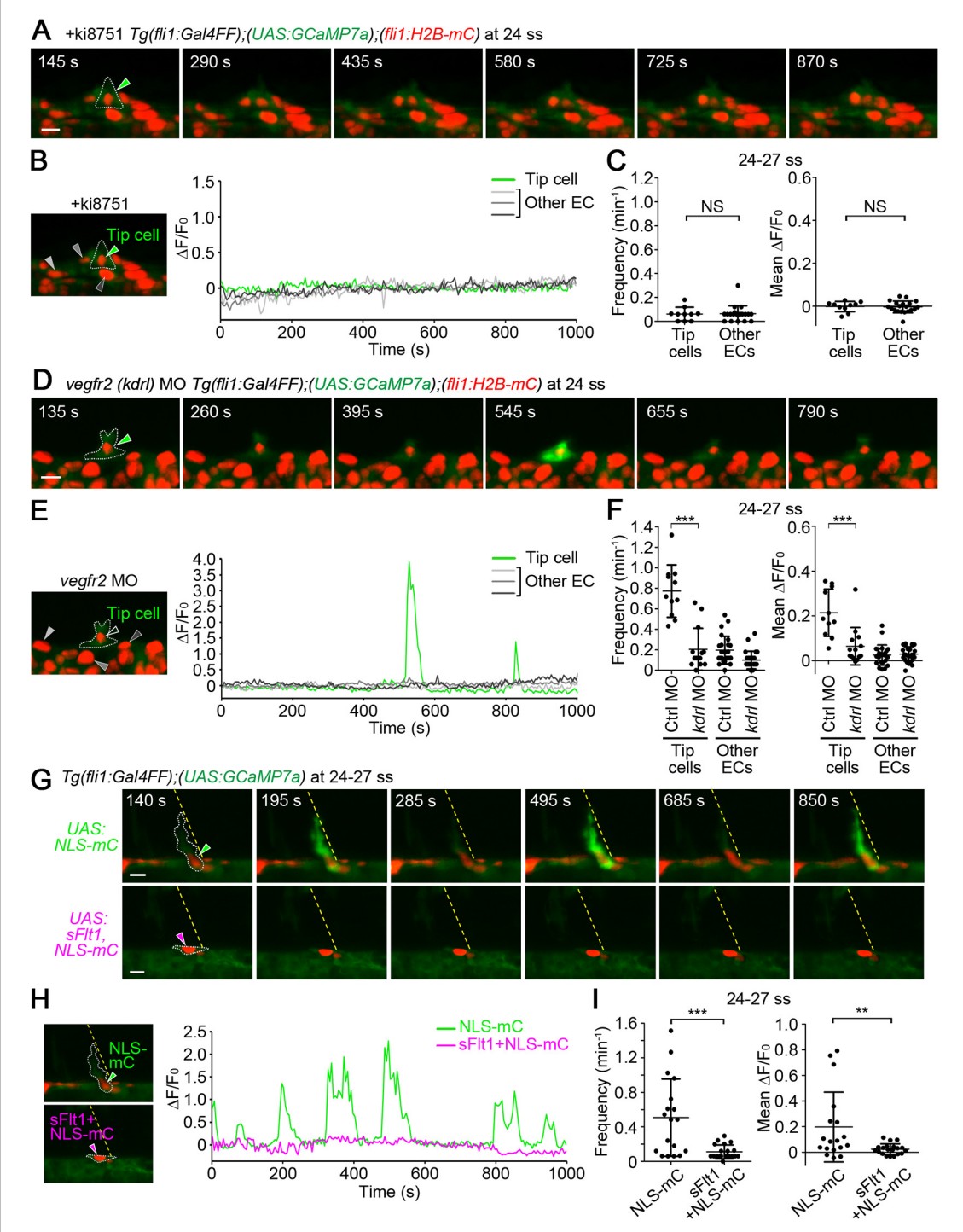

**Figure 2.** The Ca$^{2+}$ oscillations during tip cell budding depend upon Vegfa/Vegfr2 signaling. (**A**) 3D-rendered time-sequential images of *Tg(fli1: Gal4FF);(UAS:GCaMP7a);(fli1:H2B-mC)* embryos treated with a Vegfr inhibitor, ki8751, during tip cell budding. The embryos were treated from 22 ss with ki8751 and time-lapse imaged at 24 ss. A green arrowhead indicates a tip cell outlined by a dashed line. The elapsed time (s) after starting imaging of an embryo is indicated at the left upper corner. (**B**) The fluorescence changes in GCaMP7a (ΔF/F$_0$) of individual ECs from **A** indicated by arrowheads (green, light gray, dark gray, and black) at the left panel are shown as a graph. (**C**) Quantification of Ca$^{2+}$ oscillation frequency (left) and mean ΔF/F$_0$ (right) as in *Figure 1D* in ki8751-treated embryos. The embryos were treated from 22 ss with ki8751 and imaged at 24–27 ss (n ≥ 10). (**D**) 3D-rendered time-sequential images of *Tg(fli1:Gal4FF);(UAS:GCaMP7a);(fli1:H2B-mC)* embryos during tip cell budding (24 ss) injected with *vegfr2 (kdrl)* morpholino (MO). (**E**) The fluorescence changes in GCaMP7a (ΔF/F$_0$) of individual ECs from **D** indicated by arrowheads at the left panel are shown as a graph. (**F**) Quantification of Ca$^{2+}$ oscillation frequency (left) and mean ΔF/F$_0$ (right) in tip cells and other ECs within the DA in control MO- or *vegfr2*

*Figure 2 continued*

MO-injected embryos during tip cell budding at 24–27 ss (n ≥ 11). (**G**) 3D-rendered time-sequential images of *Tg(fli1:Gal4FF);(UAS:GCaMP7a)* embryos at 24–27 ss injected with control *UAS:NLS-mC* plasmid (upper) or *UAS:sFlt1,NLS-mC* plasmid (lower) which drives the expression of NLS-mC or both sFlt1 and NLS-mC simultaneously in ECs in a mosaic manner, respectively. Green and red arrowheads indicate NLS-mC-expressing ECs and both sFlt1- and NLS-mC-expressing ECs, respectively. Yellow dashed lines indicate positions of somite boundaries. (**H**) The fluorescence changes in GCaMP7a ($\Delta F/F_0$) of individual ECs from **G** indicated by arrowheads at the left panel are shown as a graph. (**I**) Quantification of $Ca^{2+}$ oscillatory activity in ECs expressing NLS-mC and both sFlt1 and NLS-mC at 24–27 ss. Graphs show $Ca^{2+}$ oscillation frequency (left) and mean $\Delta F/F_0$ (right) of NLS-mC-positive ECs within the DA close to somite boundaries (NLS-mC, n = 18; sFlt1+ NLS-mC, n = 20). Horizontal lines represent mean ± s.d.. Scale bars, 10 μm in **A**, **D** and **G**. **p < 0.01, ***p < 0.001; NS, not significant.

The following figure supplements are available for figure 2:

**Figure supplement 1.** Defects in blood vessels and lymphatic vessels found in Vegfr2- or Vegfr3-inhibited embryos.

**Figure supplement 2.** Vegfr3 is not involved in $Ca^{2+}$ oscillations in tip cells budding from the dorsal aorta (DA).

**Figure supplement 3.** $Ca^{2+}$ oscillations in the venous sprouts from the posterior cardinal vein (PCV).

(*Figure 1A,B*, *Figure 1—figure supplement 2B,C* and *Video 1*). To avoid missing the fast $Ca^{2+}$ oscillations by taking z-axis images, we performed the time-lapse 2D imaging and confirmed that $Ca^{2+}$ oscillations could be observed at more than every min (*Figure 1—figure supplement 2B,C*). In every oscillation, a $Ca^{2+}$ spike occurs throughout the cytoplasm (*Figure 1—figure supplement 2B*). The time to reach the peak of individual oscillations was varied 5.6–18.7 s (average, 9.0 s) (*Figure 1C*). Therefore, hereafter we performed 3D time-lapse imaging analyses at 5 s intervals to capture all $Ca^{2+}$ oscillations. Intracellular $Ca^{2+}$ levels of individual ECs were quantified at each time point by measuring fluorescence intensity of GCaMP7a, while tracking H2B-mC-labelled cell nuclei over time (*Figure 1—figure supplement 2D*; see Materials and methods). We analyzed $Ca^{2+}$ oscillations by the frequency and average increases in relative fluorescence intensity of GCaMP7a from the base line (mean $\Delta F/F_0$). Frequency of $Ca^{2+}$ oscillations is elevated by increased levels of agonists in some cases in ECs (*Carter et al., 1991*; *Jacob et al., 1988*; *Moccia et al., 2003*; *Mumtaz et al., 2011*) and non-ECs (*Woods et al., 1986*). Meanwhile, the amplitude of $Ca^{2+}$ rise and total $Ca^{2+}$ increases may possibly reflect the dose of agonists (*Brock et al., 1991*; *Fewtrell, 1993*; *Sage et al., 1989*). Thus, in this study, we quantified the oscillations to describe the oscillatory activity in individual EC (see 'Materials and methods'). Our quantification analyses clearly revealed that budding tip cells exhibited oscillatory activity at 24–27 ss (*Figure 1D,E*). Repetitive $Ca^{2+}$ transients were not detected in other ECs within the DA (*Figure 1A,B,D*). These results indicate that the $Ca^{2+}$ imaging method we used precisely detects the endogenous intracellular increase or decrease of $Ca^{2+}$ in vivo.

## Vegfa/Vegfr2 signaling, but not Vegfr3 signaling, is responsible for $Ca^{2+}$ oscillations in ECs sprouting from the DA

Intracellular $Ca^{2+}$ oscillations are known to occur in response to physiological concentrations of agonists in vitro in many cell types (*Fewtrell, 1993*; *Woods et al., 1986*) including ECs (*Jacob et al., 1988*; *Moccia et al., 2003*; *Sage et al., 1989*), suggesting that $Ca^{2+}$ oscillations detected here may represent EC response to angiogenic stimuli. To examine which angiogenic stimuli are responsible for $Ca^{2+}$ oscillations during vessel sprouting from the DA, we first tested the involvement of Vegfr2 since VEGF-A/VEGFR2 signaling is essential for sprouting angiogenesis (*Koch and Claesson-Welsh, 2012*; *Lohela et al., 2009*) and can increase intracellular $Ca^{2+}$ in vitro (*Figure 1—figure supplement 1C*) (*Brock et al., 1991*). Firstly, we examined the effect of inhibiting Vegfr2 on $Ca^{2+}$ oscillations by using an inhibitor of VEGFR2, ki8751 (*Kubo et al., 2005*). When we treated 22 ss embryos with ki8751 and performed $Ca^{2+}$ imaging analyses at 24–27 ss, the $Ca^{2+}$ oscillations detected in budding tip cells of control embryos (*Figure 1*) were completely abolished by the treatment of ki8751 (*Figure 2A–C*). Tip cell migration stopped in ki8751-treated embryos (*Figure 2—figure supplement 1A*), confirming that ki8751 inhibits Vegfr2 of zebrafish. ki8751 might also inhibit Vegfr3 activity, because, in our previous results, ki8751 treatment blocked Vegfr3-dependent venous sprout from the PCV in zebrafish (*Kwon et al., 2013*). Therefore, we regarded ki8751 as a zebrafish Vegfr

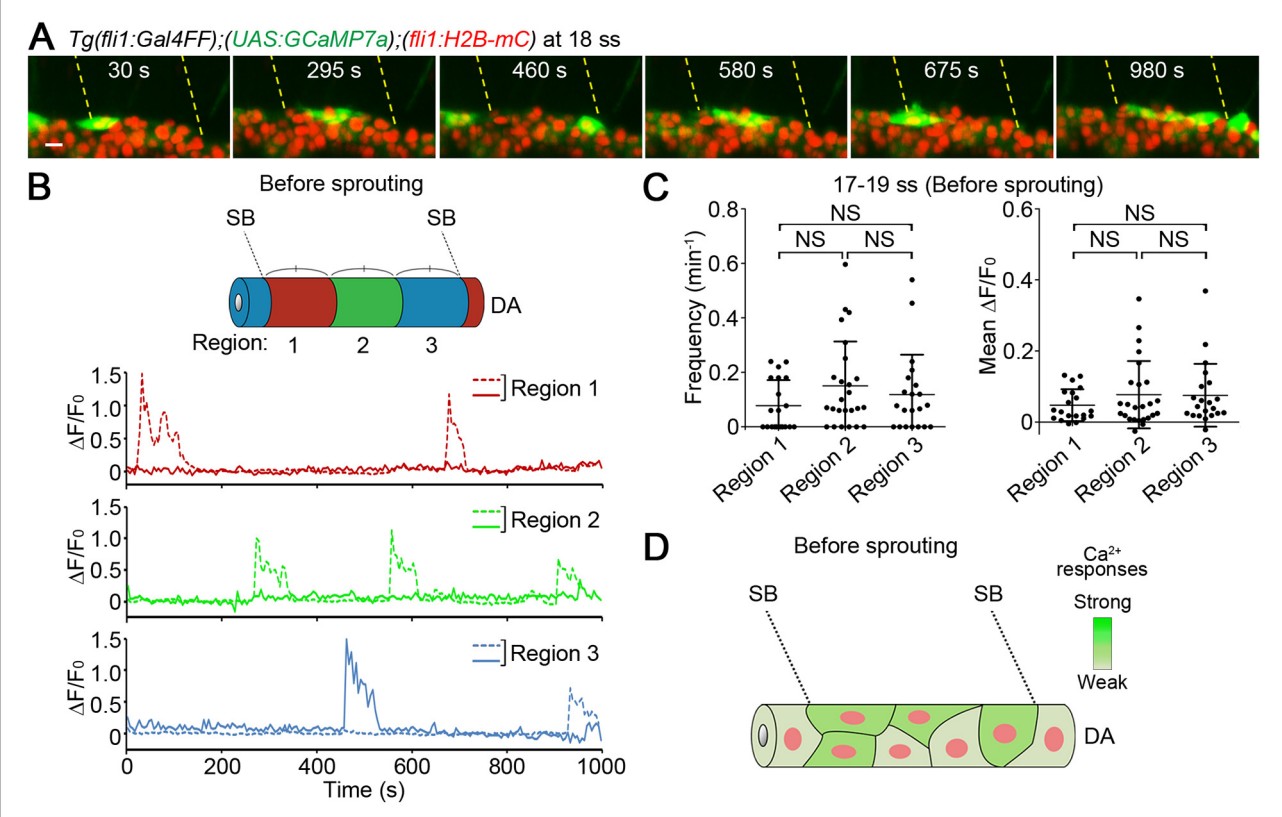

**Figure 3.** $Ca^{2+}$-oscillating cells were not restricted to specific regions within the DA before vessel sprouting. (A) 3D-rendered time-sequential images of *Tg(fli1:Gal4FF);(UAS:GCaMP7a);(fli1:H2B-mC)* embryos before ISV sprouting (18 ss). Yellow dashed lines indicate positions of somite boundaries. (B) The DA is subdivided into three regions (Region 1–3) between two somite boundaries (SBs) as illustrated in the scheme (upper). The fluorescence changes in GCaMP7a ($\Delta F/F_0$) of individual ECs from **A** are shown as separate graphs (Region 1–3), determining the region to which individual EC belongs by the location based on the position of the nucleus at the start of time-lapse imaging. A representative graph of two ECs at each region is shown. (C) Quantification of $Ca^{2+}$ oscillation frequency (left) and mean $\Delta F/F_0$ (right) in ECs of the indicated regions within the DA before vessel sprouting (17–19 ss). Horizontal lines represent mean ± s.d. (n ≥ 20). (D) Schematic illustration of $Ca^{2+}$ dynamics before tip cell budding. Before tip cells sprout from the DA, $Ca^{2+}$ oscillations are found broadly in ECs within the DA. Scale bar, 10 µm in **A**. NS, not significant. SB, somite boundary; DA, dorsal aorta.

The following figure supplement is available for figure 3:

**Figure supplement 1.** Vegfr2 is responsible for $Ca^{2+}$ responses before ISVs sprouting from the DA.

inhibitor in this study. To confirm the specific contribution of Vegfr2 in budding ECs from the DA to $Ca^{2+}$ oscillation there, we knocked-down *vegfr2* (also termed *kdrl*) by injecting antisense oligonucleotides (MO) (*Wiley et al., 2011*). Although we observed partial sprouts from the DA in the morphants as reported in *vegfr2* mutants (*Covassin et al., 2006*), $Ca^{2+}$ oscillations were markedly reduced in these budding cells (*Figure 2D–F*), suggesting that $Ca^{2+}$ oscillations in ECs budding from the DA are dependent upon Vegfr2 activation.

We further tried to confirm whether $Ca^{2+}$ oscillations depend upon Vegfa/Vegfr2 signaling by examining the effect of forced expression of soluble Flt1 (sFlt1), a potent Vegfa trap, on $Ca^{2+}$ oscillations. sFlt1 was transiently and specifically expressed in ECs in a mosaic manner by injecting plasmids at one-cell stage. While expression of control NLS-tagged mCherry (NLS-mC) did not affect $Ca^{2+}$ oscillations in budding ECs (*Figure 2G–I*; NLS-mC), co-expression of sFlt1 and NLS-mC in ECs completely inhibited $Ca^{2+}$ oscillations of ECs close to somite boundaries (*Figure 2G–I*; sFlt1+NLS-mC). Tip cell sprouting was also inhibited by expressing sFlt1 (*Figure 2—figure supplement 1B*), as previously reported (*Zygmunt et al., 2011*). Collectively, our results indicate that Vegfa/Vegfr2 signaling is required for the $Ca^{2+}$ oscillations during tip cell budding from the DA.

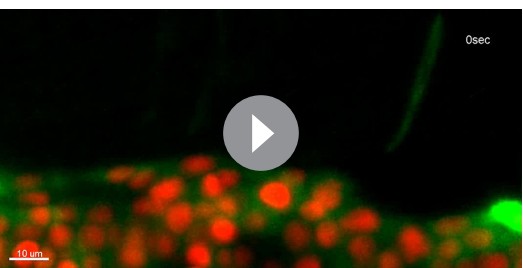

**Video 2.** Ca²⁺ oscillations occur widely within the DA before vessel sprouting. Time-lapse recording of 3D-rendered light sheet images of the *Tg(fli1:Gal4FF);(UAS:GCaMP7a);(fli1:H2B-mC)* embryos at 18 ss. Green, GCaMP7a fluorescence; red, H2B-mC fluorescence. Elapsed time is in seconds (s). Scale bar, 10 µm.

VEGFR3 signaling is another signaling pathway that facilitates sprouting angiogenesis (*Covassin et al., 2006*; *Tammela et al., 2008*). To investigate the role of Vegfr3 in Ca²⁺ oscillations in the budding ECs from the DA, we examined Ca²⁺ oscillation in *vegfr3* morphants (*Hogan et al., 2009*). *vegfr3* morphants exhibited loss of venous sprout from the PCV (*Figure 2—figure supplement 1C*) and phenocopied *vegfr3* null mutant fish (*Hogan et al., 2009*). These *vegfr3* morphants did not show any alteration of Ca²⁺ oscillations during tip cell budding (*Figure 2—figure supplement 2A–C*). These results suggest that Vegfr3 is not involved in the Ca²⁺ responses during vessel sprouting from the DA.

While ISV sprouts from the DA are mainly driven by Vegfa/Vegfr2 signaling, those from the PCV are driven by Vegfc/Vegfr3 signaling (*Hogan et al., 2009*). We observed that tip cells in the venous sprouts from the PCV exhibited Ca²⁺ oscillations in a manner dependent upon Vegfr3 (*Figure 2—figure supplement 3A–D*). Thus, Ca²⁺ oscillations are not only specific for arterial sprouting, but also occur in venous sprouting that is regulated by Vegfc/Vegfr3 signaling.

## ECs close to somite boundaries have potential to sprout from the DA

To know when and how single tip cells are selected, we looked at Ca²⁺ dynamics in ECs of the DA before ECs sprouted from the DA. Prior to sprouting from the DA, Ca²⁺ oscillations occurred widely within the DA (*Figure 3A,B* and *Video 2*) and were completely dependent upon Vegfr2 (*Figure 3—figure supplement 1A–D*). Loss of Ca²⁺ oscillations in *vegfr2* morphants (*Figure 3—figure supplement 1C,D*) was rescued in ECs expressing Vegfr2 (*Figure 3—figure supplement 1E,F*), indicating a specific role of Vegfr2 in Ca²⁺ oscillations. Although ISV sprouts usually emerge from the DA just anterior to each somite boundary (*Zygmunt et al., 2011*), Ca²⁺-oscillating cells were not restricted to specific regions within the DA before vessel sprouting (17–19 ss, *Figure 3C*). These results suggest that single tip cells are not specified among the ECs of the DA before sprouting as far as Ca²⁺ oscillation is used as an indicator (*Figure 3D*).

Next, we examined endothelial Ca²⁺ dynamics just after vessel sprouting (19–22 ss). Since ISV sprouts emerge bilaterally from the DA, we analyzed one side of sprouting ISVs. When ECs within the DA started to sprout dorsally (19–22 ss), Ca²⁺ oscillations were detected in two neighboring ECs (45.2%) or single ECs (52.4%) at somite boundaries, either of which sprouted dorsally (*Figure 4A* and *Figure 4—figure supplement 1*). We confirmed that two neighboring ECs were not the result of a recent division (86%, n = 7, data not shown). These results suggest that the initial selection of single tip cells is not completed just after the onset of vessel sprouting. Ca²⁺ response became restricted to single tip cells at later stage (24–27 ss) (*Figure 4A*), suggesting the completion of tip cell selection. Then, to visualize the process of tip cell selection between two neighboring cells exhibiting Ca²⁺ oscillations, we performed Ca²⁺ imaging for a longer period and found that only one cell maintained the oscillations over time (*Figure 4B,C* and *Video 3*). We also noticed that the ECs losing Ca²⁺ response retracted their sprout (*Figure 4B*), implying that these cells lost their ability to respond to angiogenic cues. The ECs showing sustained Ca²⁺ oscillations eventually became tip cells (18/18, *Figure 4—figure supplement 2*). Thus, our results provide evidence that the initial selection of tip cells is determined even after ECs start to migrate from the DA (*Figure 4D*).

Sema-PlexinD1 (PlxnD1) signaling is required to determine the position of tip cell budding in the vicinity of somite boundaries (*Zygmunt et al., 2011*). We observed ectopically budding ECs

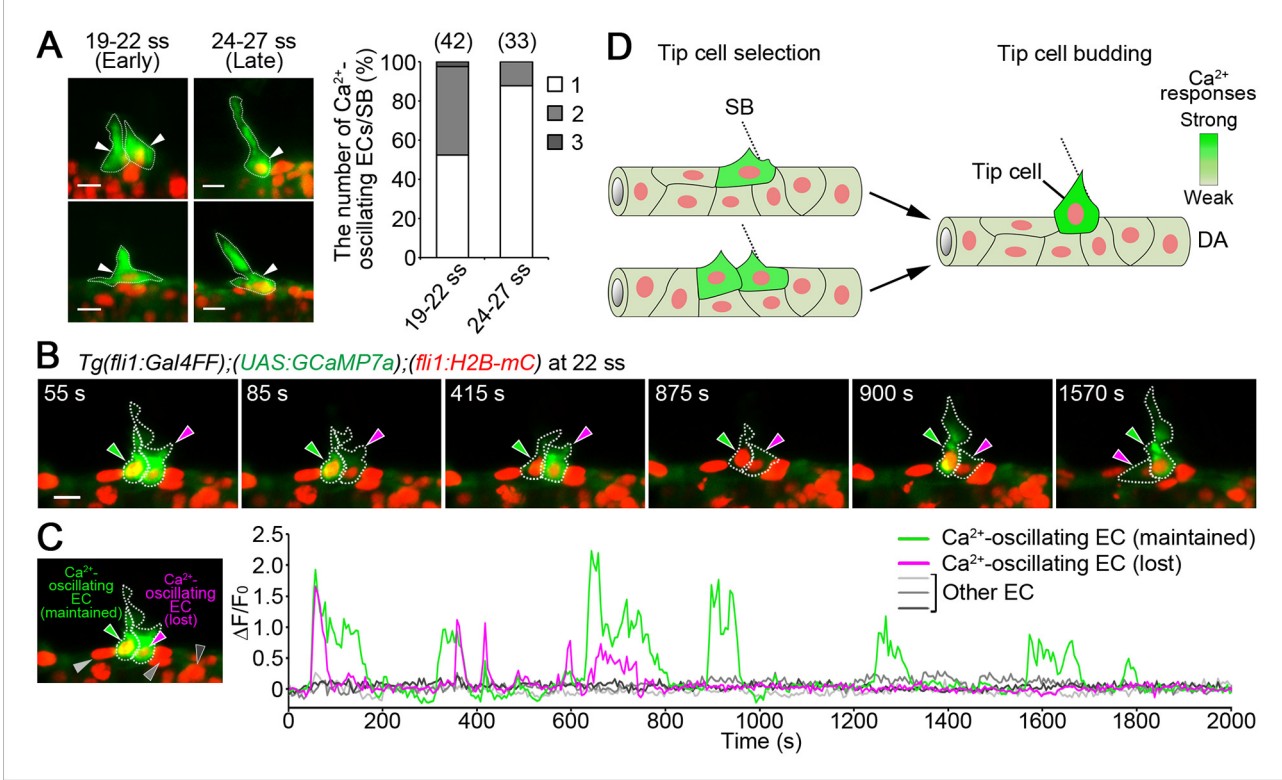

**Figure 4.** Iinitial tip cell selection in the DA. (A) The number of Ca²⁺-oscillating ECs within the DA at each somite boundary of *Tg(fli1:Gal4FF);(UAS: GCaMP7a);(fli1:H2B-mC)* embryos at 19–22 and 24–27 ss. Graph shows percentage of the number of a Ca²⁺-oscillating cell (1), two cells (2), and three cells (3) at a somite boundary among the total number of somite boundaries (indicated at the top) observed. Two each representative 3D-rendered images of *Tg(fli1:Gal4FF);(UAS:GCaMP7a);(fli1:H2B-mC)* at 19–22 and 24–27 ss are shown in the left. Arrowheads indicate Ca²⁺-oscillating cells. (B) 3D-rendered time-sequential images of *Tg(fli1:Gal4FF);(UAS:GCaMP7a);(fli1:H2B-mC)* embryos from 22 ss. Green arrowheads indicate an EC which maintained Ca²⁺ oscillations, whereas red arrowheads indicate an EC which lost Ca²⁺ oscillations. Similar results were obtained in five independent experiments. (C) The fluorescence changes in GCaMP7a (ΔF/F₀) of the ECs indicated by arrowheads in B and indicated at the left panel are shown as a graph. (D) Schematic illustration of Ca²⁺ dynamics during tip cell budding. Ca²⁺ oscillations are detected mostly in a single or two-neighboring EC(s) at the onset of vessel sprouting. Finally, only single budding tip cell exhibits Ca²⁺-oscillation at later stages. Scale bars, 10 μm in A and B. SB, somite boundary; DA, dorsal aorta.

The following figure supplements are available for figure 4:

**Figure supplement 1.** ECs close to somite boundaries have potential to sprout.

**Figure supplement 2.** Tip cell selection between two neighboring ECs exhibiting Ca²⁺ oscillations.

**Figure supplement 3.** PlexinD1 is necessary to confine Ca²⁺-oscillating sprouts in the vicinity of somite boundaries.

exhibiting Ca²⁺ oscillations in *plxnD1* morphants (*Figure 4—figure supplement 3*), confirming an essential role of PlxnD1 in the allocation of budding ECs from the DA.

## Stalk cells also exhibit Ca²⁺ oscillations during budding from the DA

We then examined Ca²⁺ dynamics at later stages of ISV formation. Sustained repetitive Ca²⁺ oscillations in tip cells were observed when ECs following tip cells were budding from the DA (*Figure 5A–C*). We defined those following ECs as stalk cells, because they eventually became stalk cells. Interestingly, Ca²⁺ oscillations also occurred in budding stalk cells (*Figure 5A,B* and *Video 4*), although their oscillation frequency and mean ΔF/F₀ were lower than those in tip cells (*Figure 5C,D*). We further found that the Ca²⁺ oscillations both in tip cells and budding stalk cells were Vegfr-dependent

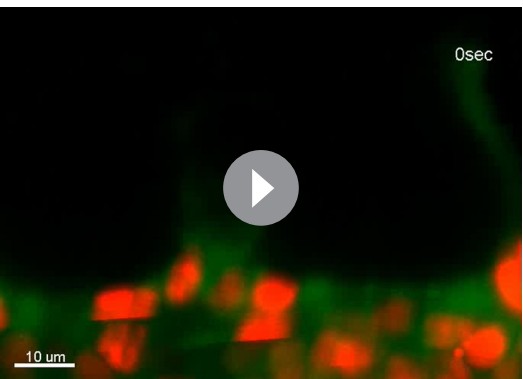

**Video 3.** The process of tip cell selection. Time-lapse recording of 3D-rendered light sheet images of the *Tg(fli1: Gal4FF);(UAS:GCaMP7a);(fli1:H2B-mC)* embryos just after vessel sprouting from the DA (22 ss). Elapsed time is in seconds (s). Scale bar, 10 μm.

(*Figure 5C* and *Figure 5—figure supplement 1*). In clear contrast, we did not detect Ca$^{2+}$ oscillations in other ECs that remained in the DA (*Figure 5A–C*).

## Vegfr2 in tip cells and stalk cells is required for their migration from the DA

To investigate the significance of the Vegfr2 activity in tip and stalk cells, we examined the behavior of ECs expressing dominant-negative Vegfr2. We generated a dominant-negative Vegfr2 by deleting the C-terminus of zebrafish Vegfr2 (Vegfr2-ΔC) (*Millauer et al., 1994*). Firstly, to validate whether Vegfr2-ΔC can block Vegfa/Vegfr2 signaling in zebrafish ECs, we established a Tg fish line, *Tg(fli1: Gal4ff);(UAS:Vegfr2-ΔC,NLS-mC),* that expresses Vegfr2-ΔC simultaneously with NLS-mC in ECs. In primary culture cells dissociated from the Tg embryos, Vegfaa-induced Erk phosphorylation was blocked in the Vegfr2-ΔC-expressing ECs (*Figure 6A,B*; Vegfr2-ΔC+NLS-mC). In contrast, in the cells dissociated from the control *Tg(fli1:Gal4ff);(UAS:NLS-mC)* embryos that expressed NLS-mC alone in ECs, the Erk phosphorylation was not blocked (*Figure 6A,B*; NLS-mC). These results confirmed that Vegfaa/Vegfr2 signaling is blocked by Vegfr2-ΔC expression. Consistently, ISV sprouting was

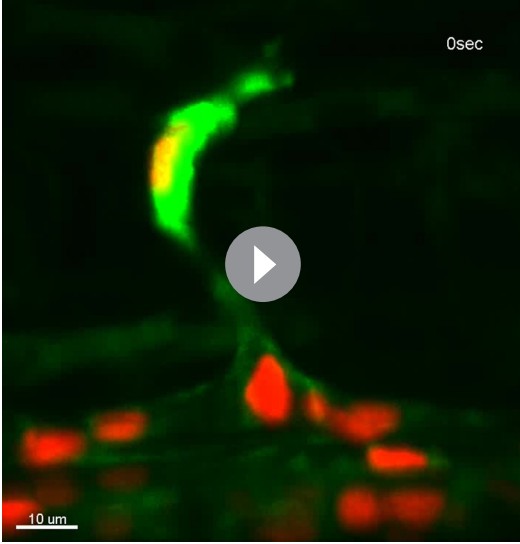

**Video 4.** Ca$^{2+}$ oscillations occur in stalk cell during budding from the DA. Time-lapse recording of 3D-rendered light sheet images of the *Tg(fli1:Gal4FF);(UAS:GCaMP7a);(fli1:H2B-mC)* embryos at 29 ss. Elapsed time is in seconds (s). Scale bar, 10 μm.

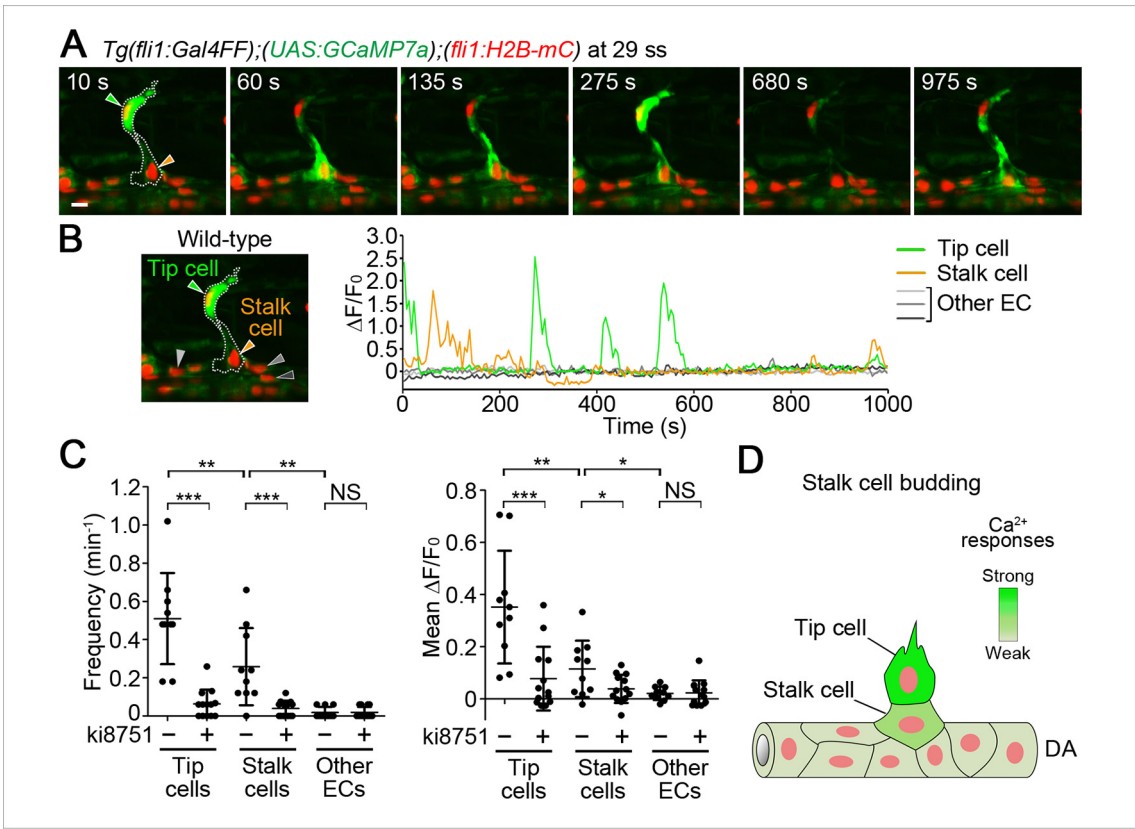

**Figure 5.** Ca$^{2+}$ oscillations in stalk cells during budding from the DA. (A) 3D-rendered time-sequential images of *Tg(fli1:Gal4FF);(UAS:GCaMP7a);(fli1:H2B-mC)* embryos during stalk cell budding from the DA (29 ss). Green and orange arrowheads indicate tip and stalk cells, respectively. (B) The fluorescence changes in GCaMP7a ($\Delta F/F_0$) of individual ECs from **A** indicated by arrowheads (green, orange, light gray, dark gray, and black) at the left panel are shown as a graph. (C) Quantification of Ca$^{2+}$ oscillatory activity in untreated and ki8751-treated embryos during stalk cell budding from the DA as in **A** and *Figure 5—figure supplement 1A*, respectively. Graphs show Ca$^{2+}$ oscillation frequency (left) and mean $\Delta F/F_0$ (right) in tip cells, stalk cells and other ECs within the DA in untreated and ki8751-treated embryos (Untreated, n $\geq$ 10; ki8751-treated, n $\geq$ 13). (D) Stalk cells that are budding from the DA have significant Vegfr2 activity, albeit weaker than that in tip cells. Scale bar, 10 μm in **A**. *p < 0.05, **p < 0.01, ***p < 0.001; NS, not significant. DA, dorsal aorta.
The following figure supplement is available for figure 5:

**Figure supplement 1.** Ca$^{2+}$ responses during stalk cell budding from the DA are dependent upon Vegfr.

blocked in the Tg embryos expressing Vegfr2-ΔC in ECs. We, then, observed the behavior of ECs expressing Vegfr2-ΔC in a mosaic manner by injecting the plasmids expressing Vegfr2-ΔC at one-cell stage. While expression of NLS-mC alone neither affect tip, stalk, nor DA cells (*Figure 6C,D*; NLS-mC), the cells expressing both Vegfr2-ΔC and NLS-mC rarely became tip and stalk cells and remained in the DA (*Figure 6C,D*; Vegfr2-ΔC+NLS-mC). These results suggest that the activation of Vegfr2 in tip cells and stalk cells is crucial for their exit from the DA.

## The Ca$^{2+}$-oscillatory activity in stalk cells coming out completely from the DA becomes comparable to that in tip cells

We then investigated Ca$^{2+}$ dynamics after stalk cells completely came out from the DA. Ca$^{2+}$ oscillations in stalk cells out of the DA were comparable to those in tip cells (*Figure 7A–C* and *Video 5*). The Ca$^{2+}$ oscillations were also dependent upon Vegfr2 (*Figure 7C* and *Figure 7—figure supplement 1*).

Ca$^{2+}$ increases generally occur by two modes: one is intracellular Ca$^{2+}$ waves which only propagate within a single cell; the other is intercellular Ca$^{2+}$ waves (ICWs) which transmit from one cell to the next via gap junctions or extracellular messengers. To investigate whether Ca$^{2+}$ oscillations were

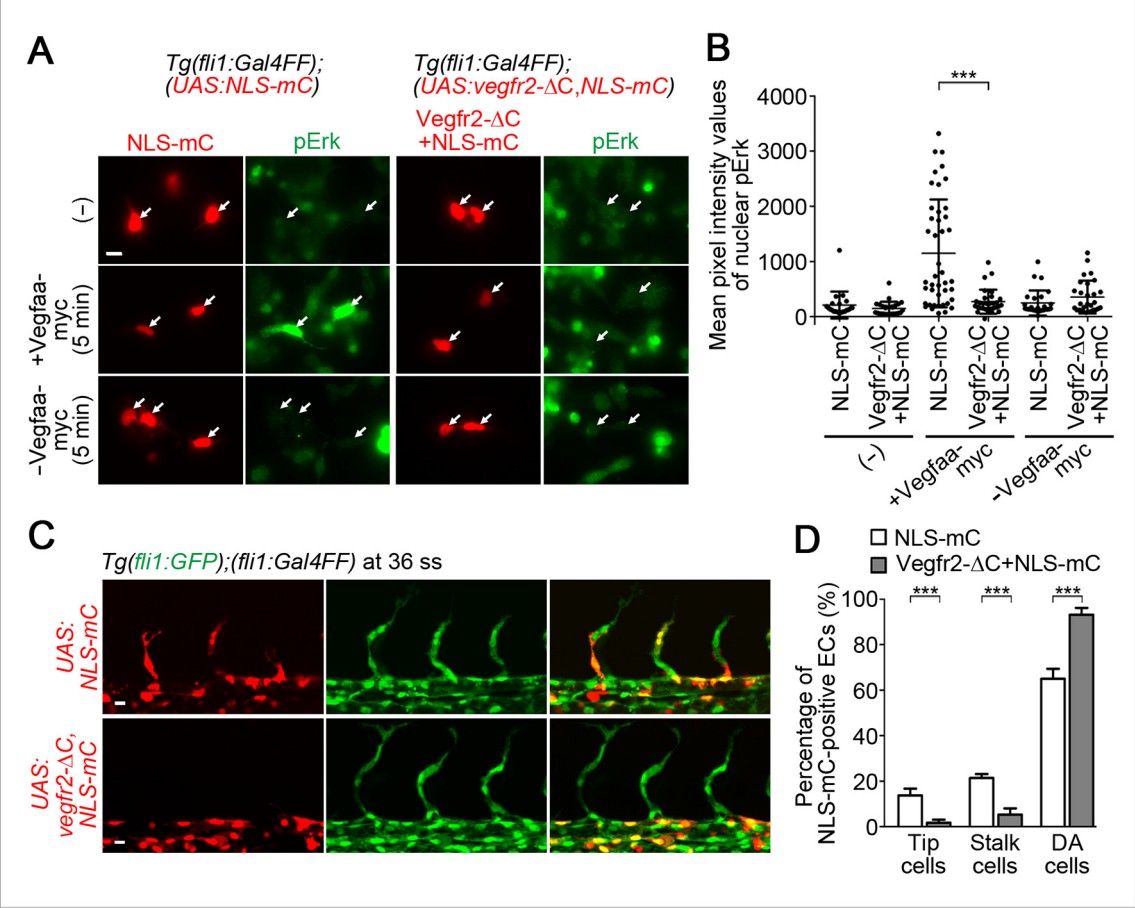

**Figure 6.** Vegfr2 activation in stalk cells is crucial for their migration from the DA. (**A**) The cells dissociated from *Tg(fli1:Gal4FF);(UAS:NLS-mC)* or *Tg(fli1: Gal4FF);(UAS:Vegfr2-ΔC,NLS-mC)* embryos (34 hpf) cultured on laminin-coated dish were kept untreated (-) or treated for 5 min with the supernatants from HEK293T cells transfected with (+) or without (-) Vegfaa-myc. The cells were immunostained with anti-phospho-Erk (pErk) antibody. mC images (red) and pErk images (green) are shown. Arrows indicate NLS-mC-positive ECs. (**B**) Quantitative analyses by the results of **A** are shown as dot-plot graphs depicting mean pixel intensity values with ± s.d. of nuclear pErk in NLS-mC-positive ECs. Each dot represents the value of single cell (n > 20). Similar results were obtained in four independent experiments. (**C**) Confocal stack fluorescence images of *Tg(fli1:GFP);(fli1:Gal4FF)* embryos at 36 ss injected with control *UAS:NLS-mC* plasmid (upper) or *UAS:Vegfr2-ΔC,NLS-mC* plasmid (lower) which drives the expression of NLS-mC or both Vegfr2-Δ C and NLS-mC simultaneously in ECs in a mosaic manner, respectively. (**D**) By counting the numbers of NLS-mC-positive ECs constituting tip cells, stalk cells, and DA cells as observed in **C** in an embryo, the percentage of each group among total number of NLS-mC-positive ECs is indicated. The data are derived from five independent experiments, in each of which ≥ 26 NLS-mC-positive cells were measured. Scale bars, 10 μm in **A** and **C**. ***p < 0.001.

due to either mechanism, we measured how often the oscillations in tip or stalk cells were synchronized with adjacent stalk or tip cells, respectively. We found that 78.9% of $Ca^{2+}$ oscillations in stalk cells occurred independently of tip cells, and 91.6% of oscillations in tip cells occurred independently of stalk cells (*Figure 7D*). These results clearly indicate that most $Ca^{2+}$ oscillations observed in tip and stalk cells occurs independently of each other and not through ICWs. Therefore, our findings suggest that not only tip cells but also stalk cells have potential to respond to angiogenic cues (*Figure 7E*). Similarly to the earlier stage, $Ca^{2+}$ oscillations were not detected in the ECs within the DA (*Figure 7A–C*), except in ECs that were budding from the DA and following stalk cells (*Figure 7—figure supplement 2*).

## Dll4/Notch signaling is crucial for suppressing $Ca^{2+}$ responses in ECs adjacent to tip cells during budding

To understand how tip cell behavior is spatio-temporally regulated by Dll4/Notch signaling, we examined the effect of the inhibition of Dll4/Notch signaling on $Ca^{2+}$ dynamics. Previous reports

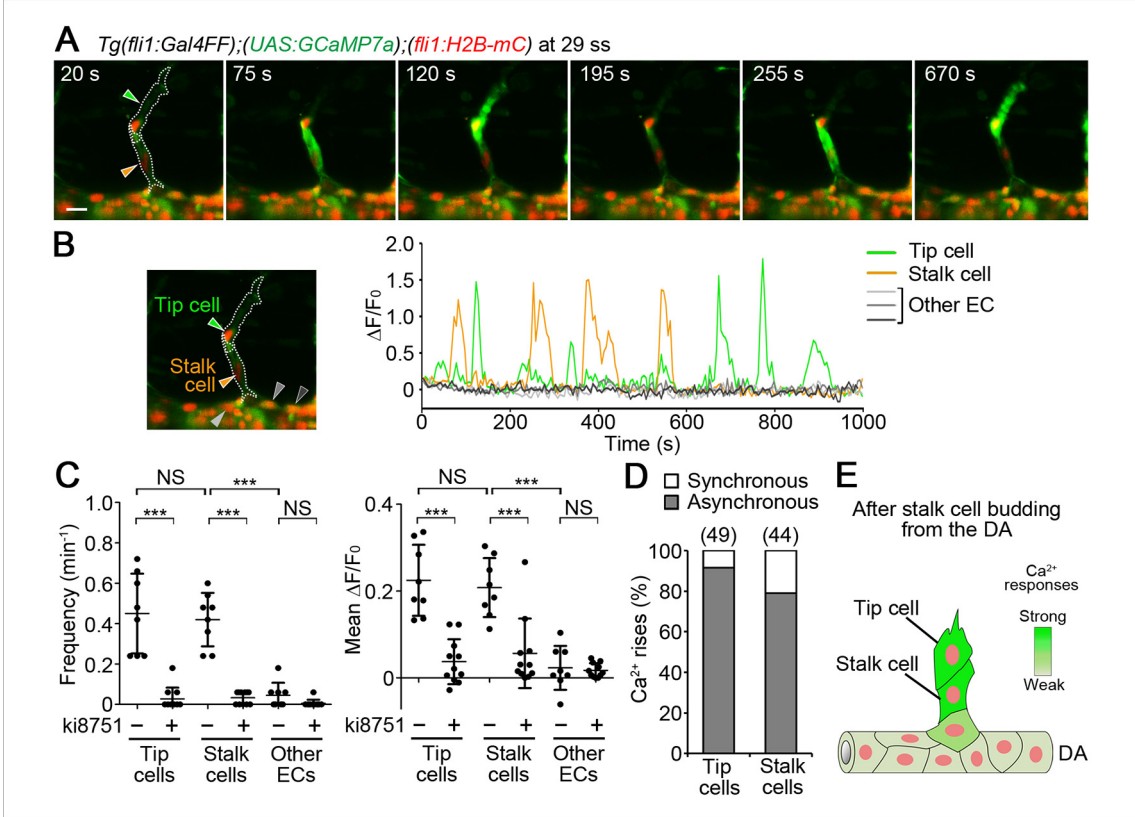

**Figure 7.** Ca$^{2+}$ oscillations in tip and stalk cells that completely come out from the DA. (**A**) 3D-rendered time-sequential images of *Tg(fli1:Gal4FF);(UAS:GCaMP7a);(fli1:H2B-mC)* embryos after stalk cells migrate out of the DA (29 ss). Green and orange arrowheads indicate tip and stalk cells, respectively. (**B**) The fluorescence changes in GCaMP7a ($\Delta F/F_0$) of individual ECs from **A** indicated by arrowheads (green, orange, light gray, dark gray, and black) at the left panel are shown as a graph. (**C**) Quantification of Ca$^{2+}$ oscillatory activity in untreated and ki8751-treated embryos after stalk cells migrate out of the DA as in **A** and *Figure 7—figure supplement 1A*, respectively. Graphs show Ca$^{2+}$ oscillation frequency (left) and mean $\Delta F/F_0$ (right) in tip cells, stalk cells and other ECs within the DA in untreated and ki8751-treated embryos. (Untreated, n ≥ 8; ki8751-treated, n ≥ 11). (**D**) Quantification of the number of synchronous and asynchronous Ca$^{2+}$ rise between tip and stalk cells. We here define the case, in which a Ca$^{2+}$ rise in one cell occurs within 10 s late behind a Ca$^{2+}$ rise in the other cell, as synchronous. The number of synchronous and asynchronous Ca$^{2+}$ rise were quantified in tip and stalk cells. Percentages of synchronous and asynchronous Ca$^{2+}$ increase to total Ca$^{2+}$ increase are shown. The total number of Ca$^{2+}$ rise analyzed in tip cells and stalk cells is indicated at the top. (**E**) Schematic illustration of Ca$^{2+}$-oscillatory activity in stalk cells. Frequency of Ca$^{2+}$ oscillations found in stalk cells are comparable to that in tip cells after stalk cells have completely come out from the DA. Scale bar, 10 mm in **A**. ***p < 0.001; NS, not significant. DA, dorsal aorta.

The following figure supplements are available for figure 7:

**Figure supplement 1.** Ca$^{2+}$ responses after stalk cell budding from the DA are dependent upon Vegfr.

**Figure supplement 2.** An EC following a stalk cell exhibits significant Ca$^{2+}$ oscillations.

have shown that MO-mediated knockdown of Dll4 leads to an increased number of ECs exhibiting tip cell behavior in zebrafish ISVs, which phenocopies loss of Notch signaling (*Siekmann and Lawson, 2007*). Therefore, we analyzed Ca$^{2+}$ oscillations before and during vessel sprouting in *dll4* morphants (*Siekmann and Lawson, 2007*). Before sprouting of ISVs, the number of Ca$^{2+}$-oscillating ECs and their oscillatory activity were significantly increased in the entire DA in *dll4* morphants (*Figure 8A–C*), suggesting that Dll4 suppresses the Vegfr2-dependent angiogenic responses in entire ECs of the DA. When ECs start budding at 19–22 ss, the Ca$^{2+}$ responses were restricted to single ECs in 44.5% at somite boundaries in the control morphants, whereas those were, in 20.8% in *dll4* morphants (*Figure 9A;* 19–22 ss). Most *dll4* morphants had two or more budding ECs exhibiting Ca$^{2+}$ oscillations (79.2%; *Figure 9A;* 19–22 ss). Thus, our findings support the hypothesis that Dll4 restricts the number of ECs showing tip cell behavior when they sprout from the pre-existing vessels.

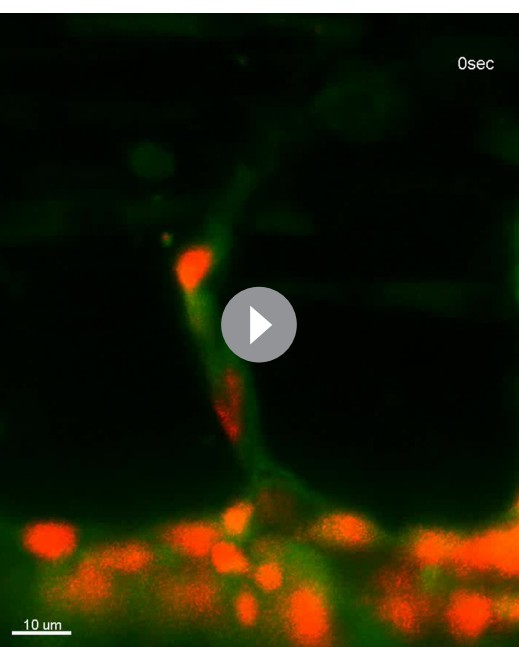

**Video 5.** Ca$^{2+}$ oscillations occur both in tip and stalk cells after stalk cell migrates out of the DA. Time-lapse recording of 3D-rendered light sheet images of the *Tg(fli1:Gal4FF);(UAS:GCaMP7a);(fli1:H2B-mC)* embryos at 29 ss. Elapsed time is in seconds (s). Scale bar, 10 µm.

In the later stages (24–27 ss), while the Ca$^{2+}$ responses became restricted to single tip cells in the control embryos, two or more neighboring ECs exhibited Ca$^{2+}$ oscillations in *dll4* morphants (74.1%; *Figure 9A;* 24–27 ss). Consistently in the *dll4* morphants, repetitive Ca$^{2+}$ oscillations were maintained in the two neighboring ECs, both of which were budding from the DA (*Figure 9B,C* and *Video 6*). Similar results were observed in *notch1b* morphants (*Figure 9—figure supplement 1*). These results suggest that Dll4/Notch signaling regulates the selection of single tip cell from two or more ECs showing Ca$^{2+}$ oscillation. In *dll4* morphants, Ca$^{2+}$ responses were detected exclusively in budding ECs, but not in other ECs within the DA (*Figure 9D*), suggesting that Dll4 suppresses the tip cell behavior especially in ECs adjacent to budding tip cells, but not in the entire DA after tip cell budding.

Next, we investigated the effect of inhibition of Vegfr in the *dll4* morphants on Ca$^{2+}$ dynamics. Treatment with ki8751 completely abolished the Ca$^{2+}$ oscillations observed in the *dll4* morphants (*Figure 9—figure supplement 2*), suggesting that an increase in the number of oscillating cells in the *dll4* morphants reflects an increase in the number of ECs showing either Vegfr2 or Vegfr3 activation. In zebrafish, arterial overproliferation and hypersprouting in the absence of Dll4/Notch signaling is, at least partially, attributed to ectopically enhanced Vegfr3 activation (*Hogan et al., 2009; Siekmann and Lawson, 2007*). Accordingly, Vegfr3 might be involved in the ectopic Ca$^{2+}$ oscillations during vessel sprouting in *dll4* morphant. To assess the contribution of Vegfr3 to the ectopic Ca$^{2+}$ oscillations, we examined Ca$^{2+}$ oscillations in *vegfr3/dll4* double morphants at 24–27 ss when the Ca$^{2+}$ responses became restricted to single tip cells in wild-type embryos (*Figure 4A*). In *vegfr3/dll4* double morphants, Ca$^{2+}$ oscillations were detected in two or more neighboring cells in 65.8% at somite boundaries (*Figure 9—figure supplement 3*), although the phenotype in the double morphants was slightly milder than that in the *dll4* morphants (74.1%; *Figure 9A*; 24–27 ss). Thus, Vegfr3 is partially involved in increases in oscillating cells in the absence of Dll4/Notch signaling. On the other hand, these results imply that ectopic Ca$^{2+}$ oscillations occur even in the absence of *vegfr3* after inhibiting Dll4/Notch signaling. Considering that the Ca$^{2+}$ responses in *dll4* morphants were completely abolished by ki8751 treatment (*Figure 9—figure supplement 2*), our findings suggest that Dll4/Notch signaling might restrict the tip cell number mainly by suppressing Vegfr2 signaling. Since Vegfr3 is responsible for hypersprouting of ISVs in the dorsal part of the *dll4* morphants

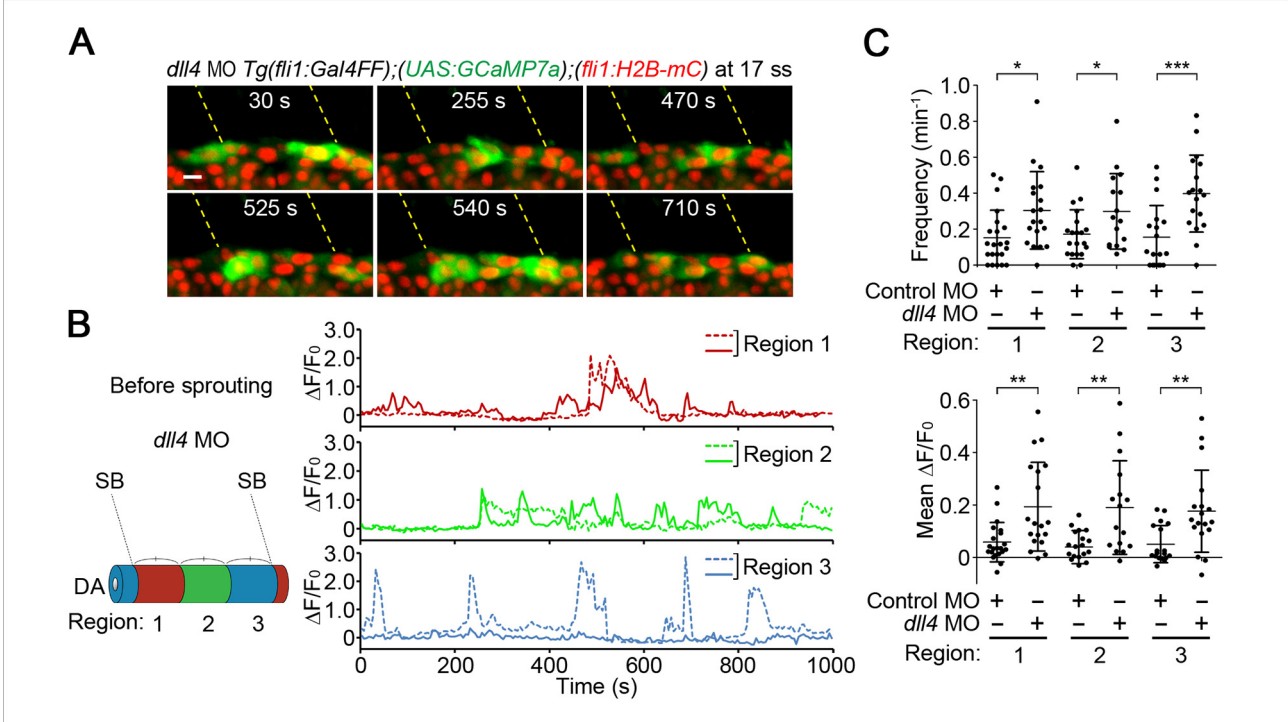

**Figure 8.** Dll4 attenuates Ca$^{2+}$ oscillations in the entire DA before ISV sprouting. (**A**) 3D-rendered time-sequential images of *Tg(fli1:Gal4FF);(UAS:GCaMP7a);(fli1:H2B-mC)* embryos before ISV sprouting injected with *dll4* MO (17 ss). Yellow dashed lines indicate positions of somite boundaries. (**B**) The DA is subdivided into three regions (Region 1–3) as illustrated in the scheme (left). The fluorescence changes in GCaMP7a (ΔF/F$_0$) of individual ECs from **A** are shown as separated graphs (Region 1–3) as in *Figure 3B*. A representative graph of two ECs at each region is shown. (**C**) Quantification of Ca$^{2+}$ oscillation frequency (upper) and mean ΔF/F$_0$ (lower) in ECs of the indicated regions within the DA in control MO- or *dll4* MO-injected embryos before vessel sprouting (17–19 ss). Horizontal lines represent mean ± s.d. (n ≥ 16). Scale bar, 10 mm in **A**. *p < 0.05, **p < 0.01, ***p < 0.001. DA, dorsal aorta.

(*Hogan et al., 2009*), Dll4/Notch signaling might not suppress Vegfr3 signaling in the early stage but suppress it in the later stage.

To investigate the mechanism how Dll4/Notch signaling suppresses Vegfr2 signaling in ECs of zebrafish, we tested the effect of inhibiting Dll4/Notch signaling on the expression of *vegfr2* during sprouting of the ISVs. However, we could detect any increase in the expression of neither *vegfr2* nor its paralog *kdr* in the trunk vessels of *dll4* morphants (*Figure 9—figure supplement 4*), suggesting that Dll4/Notch signaling inhibits the Vegfr2 signaling cascade rather than *vegfr2* expression.

## Dll4 is required for the selection of single stalk cells

We further investigated the role of Dll4/Notch signaling in stalk cell selection. Of note, when stalk cells began to migrate from the DA, Ca$^{2+}$ oscillations were mainly detected only in single stalk cells in the DA, but not in adjacent cells apart from tip cells (*Figure 10A*). Because Dll4/Notch signaling regulates the selection of single tip cells (*Figure 9*) (*Eilken and Adams, 2010*; *Lohela et al., 2009*; *Phng and Gerhardt, 2009*), we hypothesized that it might also regulate the selection of single stalk cell. To test this hypothesis, we examined the effect of knockdown of Dll4 on Ca$^{2+}$ dynamics during stalk cell budding. While Ca$^{2+}$ oscillations were mostly detected in single stalk cell in the control embryos, they were often detected in two or more neighboring ECs following tip cells in *dll4* morphants (*Figure 10A–D*). Longer time-lapse imaging confirmed that both Ca$^{2+}$-oscillating stalk cell in the *dll4* morphants eventually came out from the DA to form the ISVs. These results suggest that Dll4/Notch signaling regulates the selection of single stalk cell by restricting angiogenic behavior in the ECs adjacent to the stalk cells.

We then investigated the role of Dll4/Notch signaling after stalk cell budding from the DA. Even in the absence of *dll4*, we could not detect significant increases in oscillatory activity in both tip and

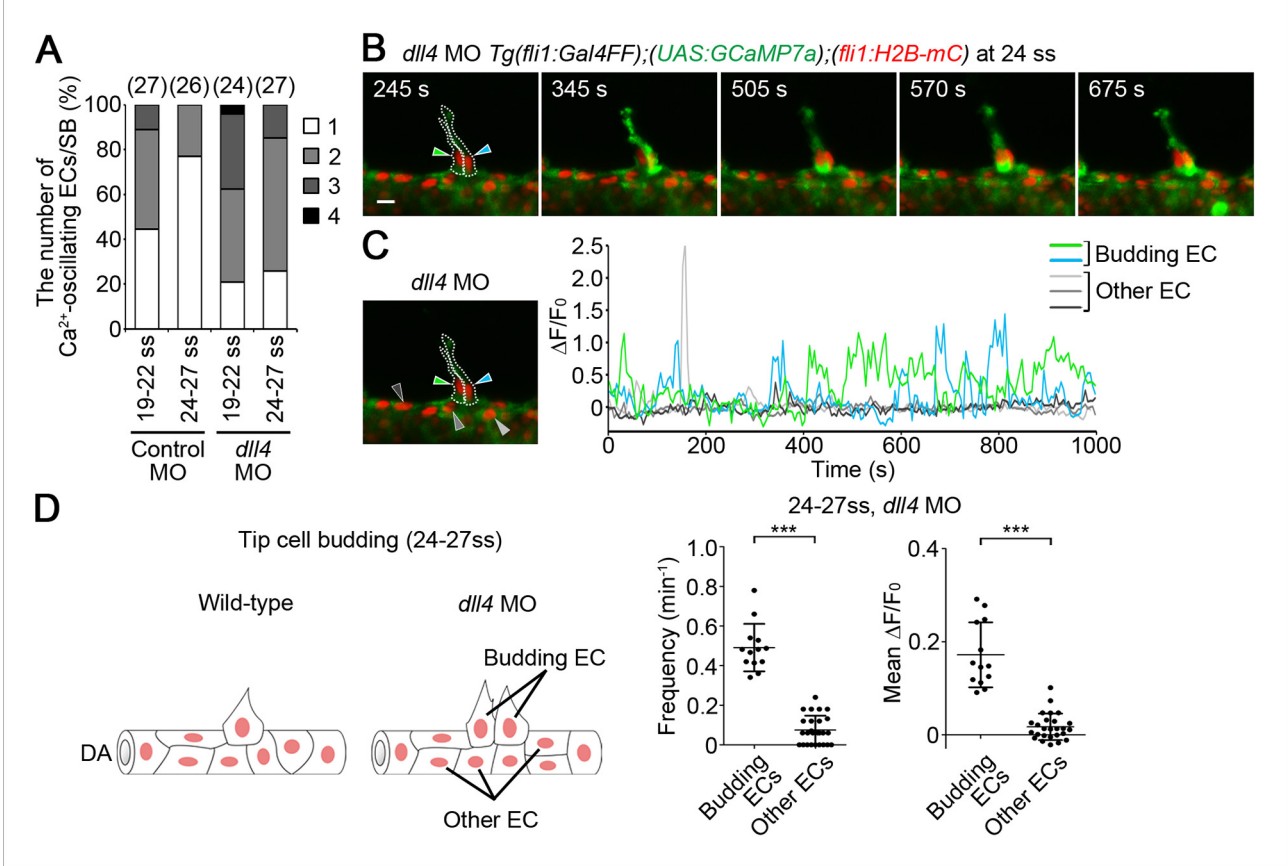

**Figure 9.** Dll4 is involved in suppressing $Ca^{2+}$ responses in ECs adjacent to tip cells. (**A**) The number of $Ca^{2+}$-oscillating ECs at each somite boundary of the embryo injected with control MO or *dll4* MO was quantified as in *Figure 4A*. (**B**) 3D-rendered time-sequential images of *Tg(fli1:Gal4FF);(UAS: GCaMP7a);(fli1:H2B-mC)* embryos during tip cell budding injected with *dll4* MO (24 ss). Green and blue arrowheads indicate two neighboring $Ca^{2+}$-oscillating ECs, both of which are budding from the DA. (**C**) The fluorescence changes in GCaMP7a ($\Delta F/F_0$) of individual ECs from **B** indicated by arrowheads at the left panel are shown as a graph. (**D**) $Ca^{2+}$ oscillation frequency (left) and mean $\Delta F/F_0$ (right) in budding ECs and other ECs within the DA in *dll4* morphants during tip cell budding at 24–27 ss as illustrated at the left panel (n ≥ 13). Scale bar, 10 mm in **B**. ***p < 0.001. DA, dorsal aorta.

The following figure supplements are available for figure 9:

**Figure supplement 1.** $Ca^{2+}$ oscillations were maintained in two neighboring ECs in *notch1b* morphants during tip cell budding from the DA.

**Figure supplement 2.** Vegfr is responsible for ectopic $Ca^{2+}$ oscillations observed in *dll4* morphants.

**Figure supplement 3.** Vegfr3 is partially involved in increases in oscillating cells in *dll4* morphants.

**Figure supplement 4.** The expression of neither vegfr2 nor kdr is not altered in the trunk vessels of dll4 morphants.

stalk cells that came out of the DA (*Figure 11A–C*). Thus, Dll4/Notch signaling does not suppress the $Ca^{2+}$-responses in the initial sprouting from the DA to form ISVs.

## Discussion

In the present study, we visualized $Ca^{2+}$ dynamics, for the first time, in ECs during sprouting angiogenesis in vivo. By quantitatively analyzing $Ca^{2+}$ dynamics in individual ECs, we have uncovered how $Ca^{2+}$ responses are spatio-temporally regulated at the single-cell level (*Figure 12*). Intracellular $Ca^{2+}$ oscillations occurred in ECs exhibiting angiogenic behavior, such as sprouting and migration. Therefore, visualizing $Ca^{2+}$ oscillations allowed us to monitor EC responses to angiogenic cues. While $Ca^{2+}$ oscillations depended upon Vegfa/Vegfr2 signaling in ECs sprouting from the DA, they were

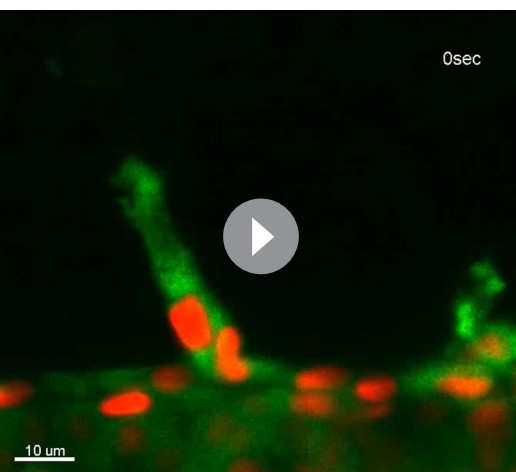

**Video 6.** Ca²⁺ oscillations are maintained in two neighboring cells in the absence of *dll4* during tip cell budding. Time-lapse recording of 3D-rendered light sheet images of the *Tg(fli1:Gal4FF);(UAS:GCaMP7a);(fli1:H2B-mC)* embryos during tip cell budding from the DA injected with *dll4* morpholino antisense oligo (MO). The recording started at 24 somite stage. Elapsed time is in seconds (s). Scale bar, 10 μm.

also detected in venous sprouts in response to Vegfc/Vegfr3. Thus, Ca²⁺ oscillations in ECs occur during both arterial and venous sprouting and subsequent migration.

Oscillatory increases of intracellular Ca²⁺ were dependent upon Vegfa/Vegfr2 signaling in ECs budding from the DA. VEGF-A treatment induces Ca²⁺ rise in cultured human ECs, although their Ca²⁺ response is not oscillatory (*Brock et al., 1991*; *Favia et al., 2014*; *Li et al., 2011*). In vitro, VEGF-A induces Ca²⁺ rise by the PLCγ-IP₃-IP₃R pathway (*Koch and Claesson-Welsh, 2012*; *Moccia et al., 2012*), by store-operated Ca²⁺ entry involving Orai1 and CRAC channel (*Li et al., 2011*) and by lysosomal Ca²⁺ release involving two-pore channel TPC2 (*Favia et al., 2014*). Since intracellular Ca²⁺ oscillations generally occur through the concerted action between several Ca²⁺ transporters (*Smedler and Uhlen, 2014*), further studies are needed to delineate whether Vegfr2 signaling directly regulates Ca²⁺ store or Ca²⁺ channels in ECs in vivo. Furthermore, it is also required to detect where intracellular Ca²⁺ increases during sprouting angiogenesis. Although we observed an increase in Ca²⁺ throughout the cytoplasm, if ECs respond to Vegfa, the initial increase of Ca²⁺ might be found in a manner dependent upon the distribution of Vegfr2. In cultured ECs, they exhibit Ca²⁺ oscillations only in the front of the cells but not in the rear during migration (*Tsai et al., 2014*). These polarized Ca²⁺ oscillations might be detected if we can improve the time/spatial resolution of imaging of zebrafish in vivo.

Ca²⁺ imaging at single-cell resolution pointed to the new regulatory mechanism underlying the formation of stalk cells following tip cells in the ISV. Lateral inhibitory action of Dll4/Notch signaling is essential for selection of tip cells (*Eilken and Adams, 2010*; *Lohela et al., 2009*; *Phng and Gerhardt, 2009*); however, it is unknown how stalk cells are specified and migrate from the parental vessels. We found that stalk cells have Vegfr-dependent Ca²⁺ oscillations when they start budding following tip cells and that Dll4/Notch signaling is required for the selection of single stalk cell similarly to tip cell selection (*Figure 12*). VEGF-A/VEGFR2 signaling is active in stalk cells and is important for cell proliferation in mouse retina angiogenesis (*Gerhardt et al., 2003*). Here, we propose that Vegfr2 activation in stalk cells is also necessary for the selection of stalk cells and their exit from the parental vessels during ISV formation. Thus, our Ca²⁺ imaging highlighted a novel role of Vegfr2 in stalk cells.

Tip cells and stalk cells are selected during sprouting angiogenesis. In the present study, we focused on the stage of ECs budding from the DA and observed that even two cells showing Ca²⁺ oscillations budded from the DA and that one of them finally became single tip cell. Jacobsson et al. report that tip cells and stalk cells change their positions in elongating vessels sprouts (*Jakobsson et al., 2010*), whereas in the earlier stage, we did not observe interexchange of tip cells

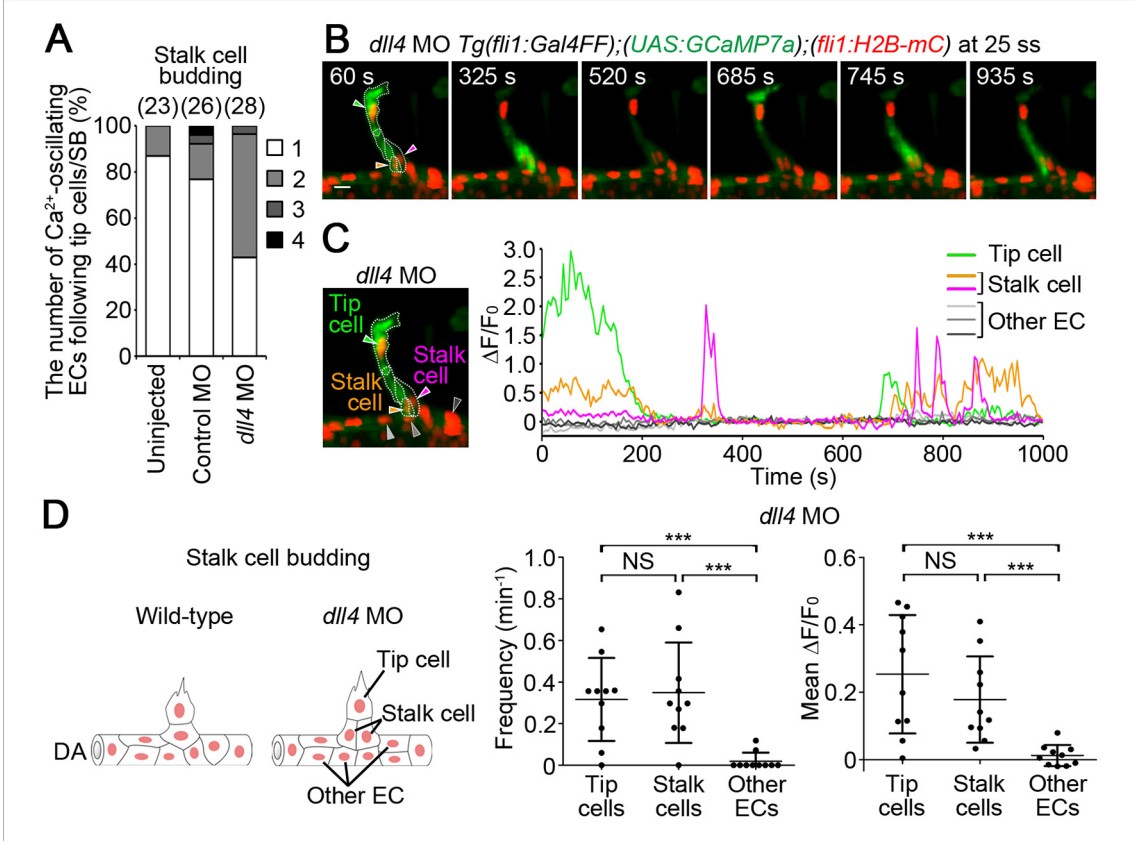

**Figure 10.** Dll4 is involved in the selection of single stalk cells. (**A**) The number of Ca$^{2+}$-oscillating ECs following tip cells in each ISV of uninjected, or control MO- or *dll4* MO-injected *Tg(fli1:Gal4FF);(UAS:GCaMP7a);(fli1:H2B-mC)* embryos during stalk cell budding when an EC/ECs following a tip cell is/are budding from the DA. Graph shows the occurrence rate of the indicated numbers of Ca$^{2+}$-oscillating cells in each ISV among the total number of ISVs observed (indicated at the top). (**B**) 3D-rendered time-sequential images of *Tg(fli1:Gal4FF);(UAS:GCaMP7a);(fli1:H2B-mC)* embryos during stalk cell budding from the DA injected with *dll4* MO (25 ss). A green arrowhead indicates tip cell. Orange and red arrowheads indicate budding stalk cells following tip cell. (**C**) The fluorescence changes in GCaMP7a (ΔF/F$_0$) of individual ECs from **B** indicated by arrowheads at the left panel are shown as a graph. (**D**) Ca$^{2+}$ oscillation frequency (left) and mean ΔF/F$_0$ (right) in tip cells, stalk cells, and other ECs within the DA in *dll4* morphants during stalk cell budding (n ≥ 10). As illustrated at the left panel, we designated budding ECs that follow tip cells as stalk cells. Scale bar, 10 mm in **B**. ***p < 0.001; NS, not significant. DA, dorsal aorta.

and stalk cells, suggesting that their plasticity of tip and stalk depends upon the stage and type of angiogenesis.

Dll4/Notch signaling is required for selection of single tip cell and single stalk cell during budding from the DA. Our findings support the notion that Dll4 in tip cells suppresses tip cell behavior in adjacent cells for the initial selection of tip cell (*Eilken and Adams, 2010*; *Herbert and Stainier, 2011*; *Lohela et al., 2009*; *Phng and Gerhardt, 2009*). Meanwhile, it has not been determined whether tip cells also suppress angiogenic responses in stalk cells via Dll4/Notch signaling during vessel elongation. Our results indicate that inhibitory activity of Dll4 toward Ca$^{2+}$ responses in stalk cells diminishes after stalk cells come out from the DA. Instead, Dll4/Notch signaling restricts the number of stalk cells. Thus, our results suggest that Dll4/Notch-mediated lateral inhibition leads to suppression of angiogenic responses mainly in ECs within the DA adjacent to budding tip or stalk cells to restrict excess budding from the DA (*Figure 12*). Our Ca$^{2+}$ imaging provide novel insights into how Dll4/Notch signaling spatio-temporally controls angiogenic behavior.

Endothelial Ca$^{2+}$ responses can be used as an indicator of cellular responses to extracellular stimuli. In this study, we show that endothelial Ca$^{2+}$ oscillations occur during sprouting angiogenesis in response to Vegfa/Vegfr2 and Vegfc/Vegfr3 signaling. We could investigate endothelial Ca$^{2+}$ responses induced by other chemical or mechanical stimuli. Among them, blood flow is a well-studied input that induces an intracellular Ca$^{2+}$ increase in ECs (*Ando and Yamamoto, 2013*). Blood

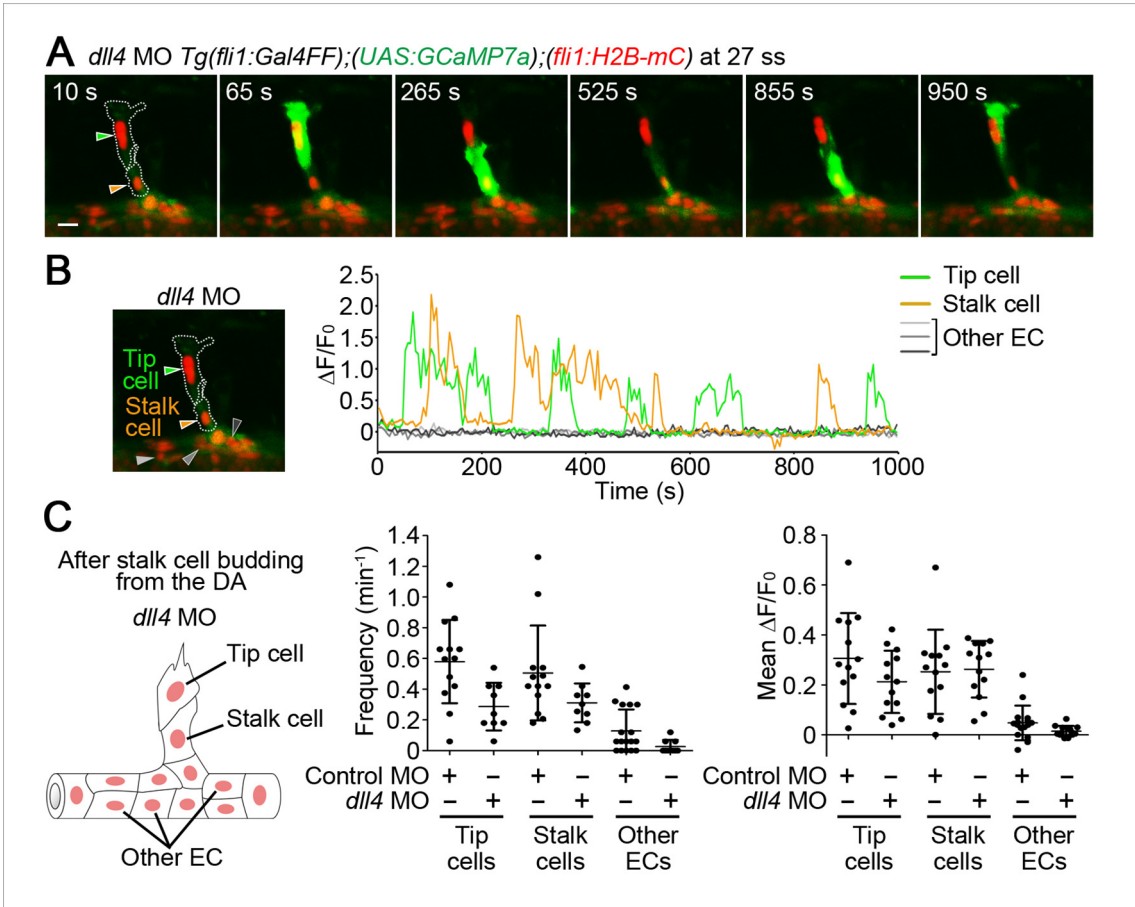

**Figure 11.** Dll4 does not regulate Ca$^{2+}$ oscillations in stalk cells after they completely migrate out of the DA. (**A**) 3D-rendered time-sequential images of *Tg(fli1:Gal4FF);(UAS:GCaMP7a);(fli1:H2B-mC)* embryos after stalk cell budding from the DA injected with *dll4* MO (27 ss). Green and orange arrowheads indicate tip and stalk cells, respectively. (**B**) The fluorescence changes in GCaMP7a ($\Delta F/F_0$) of individual ECs from **A** indicated by arrowheads at the left panel are shown as a graph. (**C**) Ca$^{2+}$ oscillation frequency (left) and mean $\Delta F/F_0$ (right) in tip cells, stalk cells, and other ECs within the DA in control MO- or *dll4* MO-injected embryos after stalk cell budding from the DA as illustrated at the left panel. (n $\geq$ 9). Scale bar, 10 mm in **A**.

flow-dependent Ca$^{2+}$ increases were recently reported in zebrafish embryos (*Goetz et al., 2014*). Our Ca$^{2+}$ imaging analyses have captured flow-dependent Ca$^{2+}$ oscillations in various types of vessels (Yokota et al., unpublished data): therefore, Ca$^{2+}$ imaging analysis will be a useful system for understanding quantitatively how individual ECs are regulated by blood flow in vivo. Thus, in vivo Ca$^{2+}$ imaging in ECs presented here will become a powerful tool to investigate the dynamic behavior of individual ECs in vascular development and homeostasis in the future studies.

## Materials and methods

### Zebrafish husbandry

Zebrafish (*Danio rerio*) were maintained and bred under standard conditions. The experiments using zebrafish were approved by the animal committee of National Cerebral and Cardiovascular Center (No. 14005) and performed according to the guidance of the Institute.

### Plasmid constructs

cDNA fragments encoding zebrafish H2B, sFlt1, Vegfr2 (Kdrl), Kdr, and Vegfaa were amplified by PCR using a cDNA library derived from zebrafish embryos and cloned into pCR4 Blunt TOPO vector (Invitrogen, Carlsbad, CA).

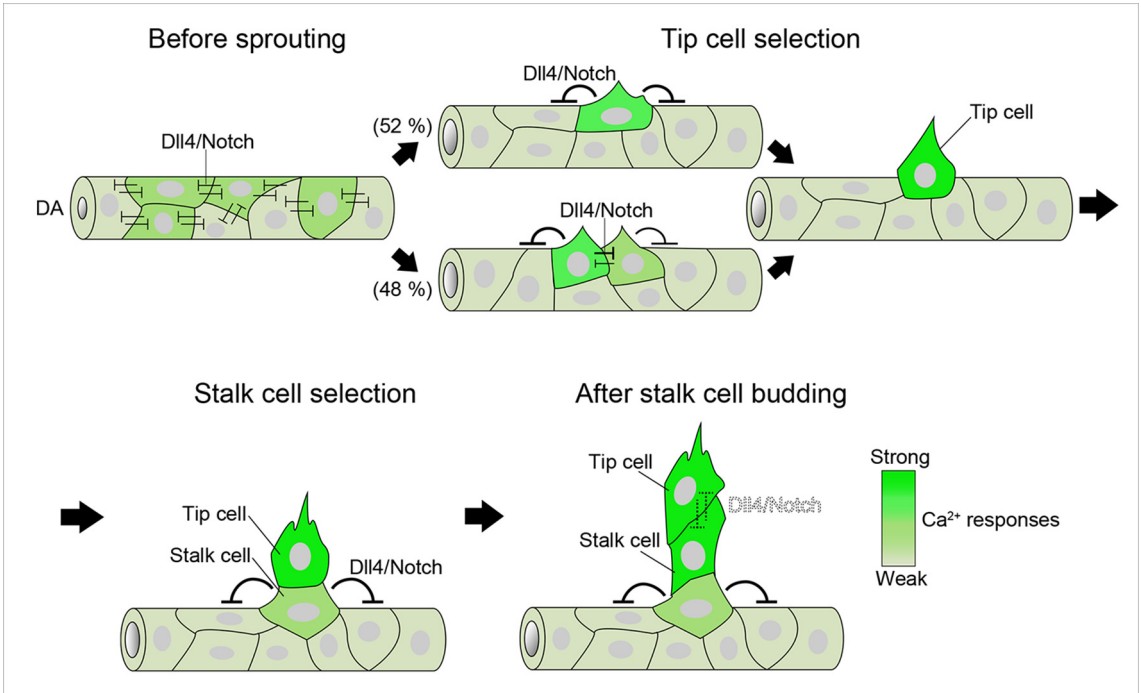

**Figure 12.** A schematic representation of endothelial $Ca^{2+}$ responses during sprouting angiogenesis from the DA. $Ca^{2+}$ oscillations occur widely within the DA before ISV sprouting. Dll4/Notch signaling attenuates $Ca^{2+}$ responses in the entire DA at this stage. $Ca^{2+}$ oscillations are restricted to single or two neighboring ECs just after vessel sprouting and then further restricted to single ECs that eventually become tip cells. Dll4/Notch signaling is required for the selection of single tip cells. During stalk cell budding, tip cells and stalk cells exhibit $Ca^{2+}$ oscillations, although the $Ca^{2+}$-oscillatory activity in stalk cells is weaker than that in tip cells. Dll4/Notch signaling regulates the selection of single stalk cells. After stalk cells completely come out from the DA, strong $Ca^{2+}$ oscillations occur both in tip cells and stalk cells. Intensity of green reflects the frequency of $Ca^{2+}$ oscillations. DA, dorsal aorta.

A cDNA encoding mCherry was subcloned into the pTol2-fli1 vector to construct the pTol2-fli1: mC plasmid (*Wakayama et al., 2015*). Then, the cDNA encoding H2B was inserted into the pTol2-fli1-mC vector to generate the pTol2-fli1:H2B-mC plasmid.

An oligonucleotide-encoding nuclear localization signal (NLS) derived from SV40 (PKKKRKV) was inserted into pmCherry-C1 vector (Clontech, Takara Bio Inc., Japan) to generate the plasmid encoding NLS-tagged mCherry (NLS-mC). The NLS-mC cDNA followed by polyA sequece was subcloned into the pTol2-E1b-UAS-E1b vector to construct the pTol2-E1b-UAS-E1b:NLS-mC plasmid (also used as the UAS:NLS-mC plasmid for microinjection) (*Wakayama et al., 2015*). The cDNA encoding zebrafish sFlt1 was inserted into the pTol2-E1b-UAS-E1b:NLS-mC vector to construct the pTol2-UAS:sFlt1,NLS-mC plasmid. The pTol2-UAS:Vegfr2-ΔC,NLS-mC was constructed by inserting a cDNA fragment that encodes Vegfr2 lacking the intracellular domain (amino acid 783–1181) into the pTol2-E1b-UAS-E1b:NLS-mC vector. The nuclear export signal (NES, derived from a HIV-1Rev protein; LQLPPLERLTLD)-tagged mC cDNA followed by polyA signal was subcloned into the pTol2-E1b-UAS-E1b vector to construct the pTol2-E1b-UAS-E1b:NES-mC plasmid. The cDNA encoding full length Vegfr2 was then inserted into the pTol2-E1b-UAS-E1b:NES-mC vector to construct the pTol2-UAS:Vegfr2,NES-mC plasmid. The cDNA encoding zebrafish Vegfaa was inserted into the pcDNA3.1 vector (Invitrogen) together with 1xMyc tag to construct the pcDNA3.1-Vegfaa-myc.

## Transgenic zebrafish lines

To generate the *Tg(fli1:H2B-mC)*, *Tg(UAS:NLS-mC)*, and *Tg(UAS:Vegfr2-ΔC,NLS-mC)* zebrafish lines, the corresponding pTol2-based plasmid DNAs (15 pg) were microinjected along with Tol2 transposase mRNA (30 pg) into one-cell stage embryos of AB or *Tg(fli1:Gal4FF)* zebrafish. Tol2 transposase mRNAs were in vitro transcribed with SP6 RNA polymerase from NotI-linearized pCS-TP vector using the mMESSAGE mMACHINE kit (Ambion, Thermo Fisher Scientific, Waltham, MA). The embryos

showing transient mCherry expression in the vasculature were selected, raised to adulthood, and crossed with wild-type AB to identify germline transmitting founder fishes.

Tg(UAS:GCaMP7a) fish and Tg(UAS:GFP);SAGFF(LF)27C fish were used (*Bussmann et al., 2010*; *Muto et al., 2013*). Tg(fli1:EGFP) fish were provided by N Lawson (University Massachusetts Medical School, USA) (*Lawson and Weinstein, 2002*). Tg(fli1:Gal4FF) fish line was a gift from M Affolter (University of Basel, Switzerland) (*Totong et al., 2011*; *Zygmunt et al., 2011*).

## Injections of morpholino oligonucleotides and plasmids

For morpholino oligonucleotide (MO)-mediated knockdown, embryos were injected at one-cell or two-cell stage with control MO (Gene Tools, LLC, Philomath, OR), 3 ng of vegfr2 MO, 7 ng of dll4 MO, 3 ng of vegfr3 MO, 3 ng of plxnD1, and 7 ng of notch1b MO. The sequences for the already-validated MOs used in this study are: vegfr2 MO, 5′-CACAAAAAGCGCACACTTACCATGT-3′;dll4 MO, 5′- TAGGGTTTAGTCTTACCTTGGTCAC-3′; vegfr3 MO, 5′-TTAGGAAAATGCGTTCT-CACCTGAG-3′; plxnD1 MO, 5′- CACACACACTCACGTTGATGATGAG-3′; notch1b MO, 5′-AATCT-CAAACTGACCTCAAACCGAC-3′ (*Leslie et al., 2007*; *Siekmann and Lawson, 2007*; *Torres-Vázquez, et al., 2004*; *Wiley et al., 2011*).

To express sFlt1, Vegfr2-ΔC, and full length Vegfr2 transiently using the Tol2 system, we co-injected the plasmids with capped Tol2 transposase mRNA (30 pg) (*Kawakami et al., 2004*). To co-express sFlt1 and NLS-mC transiently in ECs, 30 pg pTol2-UAS:sFlt1, NLS-mC (UAS:sFlt1,NLS-mC) plasmid was injected into one-cell stage of the Tg(fli1:Gal4FF);(UAS:GCaMP7a);(fli1:H2B-mC). In these embryos, the expression of sFlt1 and NLS-mC is simultaneously driven in ECs by a Gal4/UAS-based bidirectional expression system. To express both Vegfr2-ΔC and NLS-mC simultaneously in ECs, 30 pg pTol2-UAS:Vegfr2-ΔC,NLS-mC (UAS:Vegfr2-ΔC,NLS-mC) plasmid was injected into one-cell stage of the Tg(fli1:Gal4FF);(fli1:GFP). In these embryos, expression was induced in a mosaic manner (see *Figure 6C*). As a negative control, 30 pg pTol2-UAS:NLS-mC (UAS:NLS-mC) plasmid was injected. To express Vegfr2 and NES-mC simultaneously in ECs, 30 pg pTol2-UAS:Vegfr2,NES-mC (UAS:Vegfr2,NES-mC) plasmid was injected into one-cell stage of the Tg(fli1:Gal4FF);(UAS:GCaMP7a).

## Chemical treatment

The Tg(fli1:Gal4FF);(UAS:GCaMP7a);(fli1:H2B-mC) embryos were treated with 1 µM ki8751, an inhibitor of Vegfr, from 40–90 min before starting light sheet imaging, and subsequently during the imaging.

## Zebrafish image acquisition and processing

For light sheet imaging, zebrafish embryos were dechorionated and anesthetized in 0.4 mg/ml rocuronium bromide (Merk Sharp and Dohme, Kenilworth, NJ) in E3 embryo medium (5 mM NaCl, 0.17 mM KCl, 0.33 mM CaCl$_2$, 0.33 mM MgSO$_4$). The anesthetized embryos were pulled into the small glass capillaries containing 1% low-melting agarose in E3 medium, using a metal plunger. After the gel set, the fish were slightly extruded from the capillary and immersed in the chamber containing E3 medium. Light sheet imaging was performed with a Lightsheet Z.1 system (Carl Zeiss, Germany) equipped with a water immersion 20x detection objective lens (W Plan Apochromat, NA 1.0), dual sided 10x illumination objective lenses (LSMF, NA 0.2), a pco.edge scientific CMOS camera (PCO) and ZEN software. For all 3D time-lapse datasets, a z-interval of 4 µm, and a time interval of 5 s were used. z-stack images were 3D volume rendered and analyzed with IMARIS 7.7.1 software (Bitplane AG, Switzerland). For 2D time-lapse imaging, a time interval of 100 ms was used.

For confocal imaging, zebrafish embryos were dechorionated, anesthetized in 0.016% tricaine (Sigma-Aldrich, St. Louis, MO) in E3 medium, and mounted in 1% low-melting agarose poured onto a 35-mm diameter glass-based dish (Asahi Techno Glass, Japan) as previously described (*Fukuhara et al., 2014*). Confocal images were taken with a FluoView FV1000 confocal uplight microscope system (Olympus, Japan) equipped with a water-immersion 20x lens (XLUMPlanFL, NA 1.0). Images were analyzed with FV10-ASW3.1 viewer (Olympus).

## Quantitative analysis of intracellular Ca²⁺ oscillations

In our $Ca^{2+}$ imaging using light sheet microscopy, we analyzed the ISVs and DA located in middle trunk region of embryos at somite levels 8 to 14. Time-lapse images were collected every 5 s for 1000–2000 s with a z-interval of 4 µm. Z-stack images were 3D volume-rendered and analyzed with IMARIS 7.7.1 software (Bitplane AG). ISV sprouts emerge bilaterally from the DA. Thus, to focus on sprouting angiogenesis occurring on one side, our analyses were performed at either side (left or right side) of the embryos by cropping. To quantify intracellular $Ca^{2+}$ levels of individual ECs at each time point, the cell nuclei labeled with H2B-mC were automatically (or manually in some cases) tracked over time using IMARIS software. Then, for each individually tracked EC, we set a spherical region of interest (ROI) of 4–11 µm in diameters that is slightly larger than the nucleus (see *Figure 1—figure supplement 2D*). The diameters of every spherical ROI were carefully set to cover the nucleus and part of the cytoplasm but do not overlap with adjacent ECs. We then defined the highest voxel intensity of the GCaMP7a fluorescence within the ROI as the florescent intensity (F) of GCaMP7a in the EC. Because intracellular $Ca^{2+}$ waves spread rapidly and uniformly throughout the cytoplasm of each EC (see *Figure 1—figure supplement 2B*), the spherical ROIs including part of the cytoplasmic region are enough to detect intracellular $Ca^{2+}$ waves. $\Delta F/F_0$ was calculated as $(F - F_0)/F_0$, where $F_0$ is the baseline florescent intensity of GCaMP7a averaged over a 50-s period. Fluorescence changes in GCaMP7a of individual ECs are represented as $\Delta F/F_0$ traces in the graphs, and mean $\Delta F/F_0$ was calculated by taking the average of every $\Delta F/F_0$. The frequency of intracellular $Ca^{2+}$ oscillations was counted as the number of $Ca^{2+}$ oscillations per min. $\Delta F/F_0$ increase of 20% from the baseline was defined as an oscillation, while 100% is the average of the three highest $\Delta F/F_0$ peaks in oscillating ECs of wild-type embryos. The ECs, which underwent mitosis through the time-lapse images, were removed from our quantification analyses because we found that ECs never exhibit any $Ca^{2+}$ rise during M phase.

## Cell culture, transfection, and time-lapse imaging

HUVECs were purchased from Kurabo (Japan), maintained on a collagen-coated dish in endothelial cell growth medium (EGM-2, Lonza, Switzeland), and used for the experiments before passage 7. HEK293T cells were cultured in DMEM (Nacalai Tesque, Japan) with 10% FBS and antibiotics (100 mg/ml streptomycin and 100 U/ml penicillin). HUVECs and 293T cells were transfected with plasmid DNA using ViaFect transfection reagent (Promega, Madison, WI) and 293fectin transfection reagent (Invitrogen), respectively.

HUVECs expressing GCaMP7a were time-lapse imaged with an inverted fluorescence microscope (IX-81; Olympus) equipped with a Plan-Apochromat 40x/1.00 NA oil immersion objective lens (Olympus) and with a pE-1 LED excitation system (CoolLED) with a cooled charge-coupled device camera (Neo5.5 sCMOS; Andor Technology, UK). The cells were imaged at 37 °C with 5% $CO_2$ using a heating chamber (Tokai Hit, Japan).

## Preparation of zebrafish embryo primary cells, immunocytochemistry, and quantification

*Tg(fli1:Gal4FF);(UAS:NLS-mC)* or *Tg(fli1:Gal4FF);(UAS:Vegfr2-ΔC, NLS-mC)* embryos at 34 hpf were dechorionated and collected into a 1.5-ml tube after removing the yolk, washed in phosphate buffered saline (PBS), and digested with 500 µl of protease solution (PBS with 1 mg/ml trypsin, 2.7 mg/ml Collagenase P and 1mM EDTA, pH 8.0) for 15 min at 28°C under occasional pipetting. Digestion of the embryos was terminated with 50 µl of stop solution (PBS with 30% fetal bovine serum [FBS] and 6 mM calcium chloride). The dissociated cells were filtered with 35-µm cell strainers (BD Falcon, Thermo Fisher Scientific) and subsequently cultured on glass-base dish coated with laminin in L15 medium (Life Technologies, Carlsbad, CA) with 50 U/ml penicillin and 0.05 mg/ml streptomycin at 28°C. The cells spread on the culture dishes for 3.5 hr were used for experiments.

The supernatants of HEK293T cells transfected with or without the plasmid DNA encoding Vegfaa-myc were collected 36 hr after the transfection and added to zebrafish primary cells. After stimulation, the cells were fixed with 4% paraformaldehyde (PFA) in PBS for 15 min at RT, permiabilized with 0.2% Triton X-100 for 30 min at RT and blocked with 3% BSA in PBS for 30 min at RT. The cells were immunostained with anti-phospho-ERK (pERK) antibody (#4370, Cell Signaling Technology, Danvers, MA) in 3% BSA in Can Get Signal Immunostain solution (TOYOBO, Japan) at

4°C overnight. Protein reacting with the antibody was visualized with species-matched Alexa-Fluor 488-labeled secondary antibody (Invitrogen). Fluorescence images were taken with an inverted fluorescence microscope (IX-81) equipped with a Plan-Apochromat 60x/1.40 NA oil immersion objective lens, an X-cite 110LED excitation system (Excelitas Technologies, Waltham, MA) and a cooled charge-coupled device camera (CoolSNAP HQ; Roper Scientific, Photometrics, Tucson, AZ). The microscope and image acquisition were controlled by MetaMorph software (Molecular Devices, Sunnyvale, CA).

Quantification of the fluorescence intensity of pErk staining was performed with MetaMorph software. We defined a region of interest (ROI) kept at constant size (4 μm in diameter) inside the nucleus of NLS-mC-expressing ECs, and measured the mean fluorescence intensity of pErk staining within the ROI after subtracting the background.

## Whole-mount in situ hybridization

Whole-mount in situ hybridization (WISH) of zebrafish embryos was performed as described previously (*Fukuhara et al., 2014*).

## Statistical analyses

. Data were analyzed using GraphPad Prism software. Statistical significance for paired samples was determined using Student's t test. Data were considered statistically significant at $p < 0.05$.

## Acknowledgements

We thank LK Phng for critical reading for the manuscript. We are grateful to M Sone, T Babazono, W Koeda, K Hiratomi, and E Okamoto for excellent technical assistance and K Shioya for excellent fish care.

## Additional information

### Funding

| Funder | Grant reference number | Author |
| --- | --- | --- |
| Japan Society for the Promotion of Science | 22122003 | Naoki Mochizuki |
| Japan Agency for Medical Research and Development | CREST 13414779 | Naoki Mochizuki |
| Japan Society for the Promotion of Science | 15K18976 | Hiroyuki Nakajima |
| Takeda Science Foundation | | Naoki Mochizuki |
| Japan Foundation for Applied Enzymology | | Hiroyuki Nakajima |

The funders had no role in study design, data collection and interpretation, or the decision to submit the work for publication.

### Author contributions

YY, Acquisition of data, Analysis and interpretation of data; HN, Conception and design, Analysis and interpretation of data, Drafting or revising the article; YW, AM, KK, Developing transgenic zebrafish lines; SF, Conception and design; NM, Analysis and interpretation of data, Drafting or revising the article

### Ethics

Animal experimentation: The experiments using zebrafish were approved by the animal committee of National Cerebral and Cardiovascular Center (No. 14005) and performed according to the guidance of the Institute.

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
