## [Decision Letter]

Thank you for submitting your work entitled "In vivo endothelial Ca^2+^ imaging reveals spatio-temporal regulation of Vegfr2 signaling during sprouting angiogenesis" for peer review at *eLife*. Your submission has been favorably evaluated by Janet Rossant (Senior editor), a Reviewing editor, and four reviewers.

The reviewers have discussed the reviews with one another and the Reviewing editor has drafted this decision to help you prepare a revised submission.

Summary:

This paper makes use of an improved calcium activity reporter line to examine calcium dynamics at single cell resolution during sprouting angiogenesis of intersomitic vessels from the dorsal aorta in the zebrafish. Using a number of manipulations of the Vegf and Notch signaling pathways, the authors are able to provide novel information about the spatio-temporal regulation of calcium dynamics during this process.

Essential revisions:

Overall, the reviewers agreed that manuscript reflects a significant amount of work and establishes a new model to study Ca^2+^ signaling in endothelial cells in a living animal, with potentially novel insights. The experiments are carefully quantified, and the statistical analyses are well performed. Nevertheless, there are critical conceptual, as well as experimental issues that need to be addressed before this article is suitable for publication. All of the reviewers identified the same major issue of whether changes in Ca oscillations are a direct readout of VEGF signaling, as claimed, or an indirect reaction to the presence or absence of angiogenic sprouts. It was felt that the conclusions in the Results section and discussion on using GCaMP7 as a bonafide read out of VEGFR2 activity dynamics in vivo is premature and problematic.

In order to address this issue, the reviewers made the following recommendations which need to be addressed in any revised version of the manuscript:

1) The authors should tone down their conclusions that they have a live readout for VEGFR signaling and focus on the correlation with endothelial activity in migration and sprouting, rather than as a direct readout of VEGFR signaling.

2) In order to further explore whether the calcium oscillations are directly related to VEGF signaling, it is suggested that you examine whether similar oscillations occur in endothelial cells that sprout in response to Vegfc/Vegfr3 or Bmp signaling. This recommendation is essential for further consideration.

3) It is also suggested that you might attempt mosaic overexpression of Vegfr2 in ECs, in order to assess whether this gain of function could induce Ca^2+^ responses during the angiogenic process. This recommendation may be harder to perform and interpret and so is not absolutely mandatory

4) Because the transgenic reporter line is new, some control experiments are crucial for publication. The authors should show that the endothelial-specific Ca probe responds as expected by looking at well-established agonists and antagonists of Ca signaling.

---

## [Author Response]

*Essential revisions:*

*Overall, the reviewers agreed that manuscript reflects a significant amount of work and establishes a new model to study Ca^2+^ signaling in endothelial cells in a living animal, with potentially novel insights. The experiments are carefully quantified, and the statistical analyses are well performed. Nevertheless, there are critical conceptual, as well as experimental issues that need to be addressed before this article is suitable for publication. All of the reviewers identified the same major issue of whether changes in Ca oscillations are a direct readout of VEGF signaling, as claimed, or an indirect reaction to the presence or absence of angiogenic sprouts. It was felt that the conclusions in the Results section and discussion on using GCaMP7 as a bonafide read out of VEGFR2 activity dynamics in vivo is premature and problematic. In order to address this issue, the reviewers made the following recommendations which need to be addressed in any revised version of the manuscript:*

First of all, we thank the editor and the reviewers for their practical suggestions and comments to improve our manuscript. As the reviewers pointed out, our claim that endothelial Ca^2+^ oscillations are a bonafide readout of Vegfr2 signaling is premature, because Ca^2+^ oscillations are only a branch of multiple Vegfr2 downstream signaling. According to the reviewers’ advice, we weakened the conclusion by changing the title and rewriting the manuscript accordingly. We now believe that the results obtained by our new additional experiments thoroughly fulfill the reviewers’ concerns. Our replies to each point are as follows.

*1) The authors should tone down their conclusions that they have a live readout for VEGFR signaling and focus on the correlation with endothelial activity in migration and sprouting, rather than as a direct readout of VEGFR signaling.*

According to the reviewers’ advice, we have toned down our conclusions that Ca^2+^ responses are a live readout for Vegfr2 signaling and have instead highlighted the conclusions that endothelial Ca^2+^ responses correlate with angiogenic cellular behaviors such as migration and sprouting. We have focused on the Ca^2+^ oscillations as a result of responses to angiogenic cues. We have changed the title and rewritten the revised manuscript.

*2) In order to further explore whether the calcium oscillations are directly related to VEGF signaling, it is suggested that you examine whether similar oscillations occur in endothelial cells that sprout in response to Vegfc/Vegfr3 or Bmp signaling. This recommendation is essential for further consideration.*

The angiogenesis we looked at is about the sprouting from the main trunk vessels of zebrafish. Therefore, according to the reviewers’ suggestions, we examined the possibility that Vegfc/Vegfr3 signaling instead of BMP might also induce Ca^2+^ oscillations.

We have monitored Ca^2+^ responses in ECs of the venous sprouts (secondary sprouts) from the posterior cardinal vein (PCV), which occurred in response to Vegfc/Vegfr3 signaling (Hogan et al. Development 136:4001-4009, 2009). Tip cells in the venous sprouts exhibited apparent Ca^2+^ oscillations in a manner dependent on Vegfr3. These data are included as new Figure 2—figure supplement 3.

*3) It is also suggested that you might attempt mosaic overexpression of Vegfr2 in ECs, in order to assess whether this gain of function could induce Ca^2+^ responses during the angiogenic process. This recommendation may be harder to perform and interpret and so is not absolutely mandatory*

According to the reviewers’ advice, we have overexpressed Vegfr2 and NES (nuclear export signal) tagged-mCherry to mark the cells in which overexpressed Vegfr2 is distinguished from other ECs in a mosaic manner. However, we could not detect significant enhancement of Ca^2+^ oscillations in the ECs expressing exogenous Vegfr2 before or during sprouting angiogenesis from the dorsal aorta (DA). We consider that overexpression of Vegfr2 could not cause a gain of function effect, because ECs might have a surplus amount of endogenous Vegfr2 to respond to endogenous Vegfa. Then, we examined the effect of Vegfr2 over-expression in *vegfr2* morphants. Whereas Ca^2+^ oscillations were abolished in ECs within the DA in *vegfr2* morphants, exogenous Vegfr2 recovered Ca^2+^ oscillations in the morphants (new Figure 3—figure supplement 1). These new results support our notion that Vegfr2 is an essential regulator of the Ca^2+^ oscillations in the DA.

*4) Because the transgenic reporter line is new, some control experiments are crucial for publication. The authors should show that the endothelial-specific Ca probe responds as expected by looking at well-established agonists and antagonists of Ca signaling.*

Precisely following the reviewers’ comments, we have confirmed that GFP from GCaMP7 reflects changes in intracellular Ca^2+^ levels in ECs. While inonomycin treatment markedly enhanced GCaMP7a fluorescence in HUVECs (new Figure 1—figure supplement 1) and in zebrafish (new Figure 1—figure supplement 1), BAPTA-AM treatment blocked enhancement of GCaMP7a fluorescence induced by VEGF-A (new Figure 1—figure supplement 1).